# Life expectancy changes since COVID-19

**Jonas Schöley** [1] ✉, **José Manuel Aburto** [2,3,4,5] ✉, **Ilya Kashnitsky** [4],
**Maxi S. Kniffka** [1], **Luyin Zhang** [2], **Hannaliis Jaadla** [6,7], **Jennifer B. Dowd** [2,3]
and **Ridhi Kashyap** [2,3] ✉

The COVID-19 pandemic triggered an unprecedented rise in mortality that translated into life expectancy losses around the world, with only a few exceptions. We estimate life expectancy changes in 29 countries since 2020 (including most of Europe, the United States and Chile), attribute them to mortality changes by age group and compare them with historic life expectancy shocks. Our results show divergence in mortality impacts of the pandemic in 2021. While countries in western Europe experienced bounce backs from life expectancy losses of 2020, eastern Europe and the United States witnessed sustained and substantial life expectancy deficits. Life expectancy deficits during fall/winter 2021 among people ages 60+ and <60 were negatively correlated with measures of vaccination uptake across countries ($r_{60+} = -0.86$; two-tailed $P < 0.001$; 95% confidence interval, $-0.94$ to $-0.69$; $r_{<60} = -0.74$; two-tailed $P < 0.001$; 95% confidence interval, $-0.88$ to $-0.46$). In contrast to 2020, the age profile of excess mortality in 2021 was younger, with those in under-80 age groups contributing more to life expectancy losses. However, even in 2021, registered COVID-19 deaths continued to account for most life expectancy losses.

Period life expectancy (LE) is a summary measure of current population health. If mortality increases in a population, LE declines. Conversely, if mortality declines, LE increases. The measure is age-standardized and thus commonly employed for international comparisons of population health. In this paper we investigate LE changes since the start of the pandemic, distinguishing countries that saw worsening losses from countries that managed to bounce back from their LE drop in 2020.

Most countries experienced sizable gains in LE during the second half of the twentieth century[1]. However, at the turn of the twenty-first century, the rate of improvement in LE slowed in many high-income countries prior to the COVID-19 pandemic[2], such as the United States[3], England and Wales[4], and Scotland[5], among others[3]. The COVID-19 crisis triggered a mortality shock resulting in LE declines in 2020 of a magnitude not observed in the recent history of high-income countries[6–9]. While data limitations have precluded in-depth analyses in

low-to-middle-income countries, emerging evidence suggests even larger losses than those observed in high-income countries (such as in India[10] and Latin America[11–13]). Only very few countries did not witness declines in LE in 2020, including Norway, Denmark, Finland (for females only), New Zealand and Australia[6,7,14,15].

Fluctuations in LE are not uncommon. Typically, LE declines are quickly followed by bounce backs[16,17]. In contrast to these short-term fluctuations, however, the COVID-19 pandemic induced global and severe mortality shocks in 2020 and, as of spring 2022, is still ongoing. Throughout 2021, the impact of the pandemic became more heterogeneous across populations with differences in prior infection, non-pharmaceutical interventions and vaccination uptake, all influencing the pandemic's course. Emerging estimates of LE losses based on excess deaths suggest that most western European countries are expected to partly recover from the losses observed in 2020, while

[1]Max Planck Institute for Demographic Research, Rostock, Germany. [2]Leverhulme Centre for Demographic Science and Department of Sociology, University of Oxford, Oxford, UK. [3]Nuffield College, University of Oxford, Oxford, UK. [4]Interdisciplinary Centre on Population Dynamics, University of Southern Denmark, Odense, Denmark. [5]Department of Population Health, London School of Hygiene and Tropical Medicine, London, UK. [6]Estonian Institute for Population Studies, Tallinn University, Tallinn, Estonia. [7]Cambridge Group for the History of Population and Social Structure, Department of Geography, University of Cambridge, Cambridge, UK. ✉e-mail: schoeley@demogr.mpg.de; jose.aburto@lshtm.ac.uk; ridhi.kashyap@nuffield.ox.ac.uk

other countries (including the United States and Russia) will suffer further LE declines[18].

We examine LE changes since 2019 in 29 countries, including most of Europe, the United States and Chile, using data on all-cause mortality from the Short-Term Mortality Fluctuations Database[19] following a previously validated methodology[6]. We distinguish between annual LE changes and LE deficits, with the former showing the raw year-to-year difference and the latter indicating the difference between observed and expected LE had pre-pandemic trends continued. Using decomposition techniques, we describe which age groups and to what extent registered COVID-19 deaths contributed to recent trends and deficits in LE. We compare the magnitude and length of the current global LE decline with prominent mortality shocks during the twentieth century. All our results are reported for females, males and the total population. We further investigate associations between LE changes and vaccination uptake. Our results quantify the mortality burden of COVID-19 in 2021 and contribute to the debate about recent trends in LE from a cross-national perspective.

## Results

### Changes in LE since 2019

Among the 29 countries analysed, 8 countries saw significant LE bounce backs from 2020 losses: Belgium (+10.8 months; 95% confidence interval (CI), 9.7 to 11.9; $H_0: \mu \leq 0$; $P < 0.001$), Switzerland (+7.7; 95% CI, 6.4 to 8.8; $P < 0.001$), Spain (+7.6; 95% CI, 7.1 to 8.1; $P < 0.001$), France (+5.0; 95% CI, 4.4 to 5.6; $P < 0.001$), England and Wales (+2.1; 95% CI, 1.6 to 2.7; $P < 0.001$), Italy (+5.1; 95% CI, 4.6 to 5.5; $P < 0.001$), Sweden (+7.5; 95% CI, 6.0 to 8.6; $P < 0.001$) and Slovenia (+3.1; 95% CI, 0.4 to 5.7; $P = 0.010$). Compounding the 2020 losses, LE dropped significantly further throughout 2021 in 12 countries: Bulgaria (−25.1 months; 95% CI, −23.4 to −26.6; $H_0: \mu \geq 0$; $P < 0.001$), Chile (−8.0; 95% CI, −7.0 to −9.0; $P < 0.001$), the Czech Republic (−10.4; 95% CI, −9.4 to −11.5; $P < 0.001$), Germany (−3.1; 95% CI, −2.7 to −3.5; $P < 0.001$), Estonia (−21.5; 95% CI, −17.6 to −25.1; $P < 0.001$), Greece (−12.4; 95% CI, −11.0 to −13.8; $P < 0.001$), Croatia (−11.6; 95% CI, −9.7 to −13.3; $P < 0.001$), Hungary (−16.4; 95% CI, −15.3 to −17.6; $P < 0.001$), Lithuania (−7.9; 95% CI, −5.4 to −10.5; $P < 0.001$), Poland (−12.1; 95% CI, −11.3 to −12.7; $P < 0.001$), Slovakia (−23.9; 95% CI, −22.3 to −25.7; $P < 0.001$) and the United States (−2.7; 95% CI, −2.2 to −3.1; $P < 0.001$). LE in Scotland and Northern Ireland remained at approximately the same depressed levels as in 2020, indicating a constant excess mortality. In terms of LE changes since 2019, the extremes are marked by Bulgaria, with record compound LE losses across 2020 and 2021, and France, Belgium, Switzerland and Sweden, all with complete LE bounce backs from substantial prior losses. Of the three countries that experienced no LE loss in 2020 (Denmark, Norway and Finland), only Norway had a significantly higher LE in 2021 than in 2019 (Fig. 1 and Table 1).

In all countries, LE in 2021 was lower than expected under the continuation of pre-pandemic trends. Bulgaria, Chile, Croatia, the Czech Republic, Estonia, Germany, Greece, Hungary, Lithuania, Poland and Slovakia suffered substantially higher LE deficits in 2021 than in 2020, indicating a worsening mortality burden over the course of the pandemic (Extended Data Figs. 1–3).

Bulgaria experienced 17.8 months of LE decline in 2020 (95% CI, −16.5 to −19.8; $H_0: \mu \geq 0$; $P < 0.001$). This substantial decline was compounded by an even larger loss of 25.1 months (95% CI, −23.4 to −26.6; $P < 0.001$) below the 2020 level in 2021, leaving the country with a net LE loss of 43.0 months (95% CI, −41.4 to −44.5; $P < 0.001$) since 2019. Bulgaria is the most severe example regarding LE losses among the nine countries from the former Eastern Bloc (Bulgaria, Slovakia, Lithuania, Poland, Estonia, Hungary, the Czech Republic, Croatia and Slovenia). Except Slovenia, all these countries suffered compound LE losses in 2021. Estonia stands out as the country with the third-largest LE losses in 2021 but almost no losses in 2020. Substantial compound losses were also observed in Chile and Greece. In contrast, after an 8-month

LE loss in 2020, Switzerland experienced a bounce back of 7.7 months (95% CI, 6.4 to 8.8; $H_0: \mu \leq 0$; $P < 0.001$). Belgium, Sweden, France, Italy and Spain joined Switzerland as countries that witnessed bounce backs from substantial LE losses in 2020, with the first three countries having regained the LE levels of 2019 (Fig. 1 and Table 1).

### Age contributions to LE changes

In 2021, the pandemic death toll shifted towards younger age groups. For example, while US mortality for ages 80+ returned to pre-pandemic levels in 2021, overall LE losses grew due to worsening mortality in ages below 60. Mortality increases among ages below 60 contributed LE losses of −7.2 months (95% CI, −7.0 to −7.4; $H_0: \mu \geq 0$; $P < 0.001$) in 2021 compared with 2020. These LE losses among the young cancel the LE bounce backs among the older population and yield a net LE drop in 2021 of −2.7 months (95% CI, −2.2 to −3.1; $H_0: \mu \geq 0$; $P < 0.001$) for the United States (Fig. 2). Excess mortality among under-60s explained more than half of the loss in US LE since the start of the pandemic (58.9%; 95% CI, 57.9 to 59.8; $H_0: \mu \leq 50\%$; $P < 0.001$). LE losses in the under-60s, especially for males, were considerably higher in the United States than in most other countries in 2020 as well[6].

The pattern of the shift in excess mortality away from the oldest ages in 2021 compared with 2020 is also evident in Austria, Belgium, the Czech Republic, England and Wales, Germany, the Netherlands, Northern Ireland, Poland, Portugal, Scotland, Slovakia, Slovenia and Spain (Fig. 2). In 11 of the 16 countries with LE losses in 2021, the under-60 age groups contributed significantly more to LE loss in 2021 than in 2020 (one-tailed, $P < 0.05$). Among the 13 countries that partially or completely bounced back from their LE losses in 2020, 10 (Austria, Belgium, Switzerland, Spain, France, England and Wales, Italy, Netherlands, Sweden and Slovenia) achieved the bounce back primarily or solely due to normalizing mortality among the older population ($H_0: \mu \leq 50\%$ contribution of ages 60+ to LE change; $P < 0.05$; Table 1). Croatia, Greece, Hungary, Northern Ireland and Slovakia saw almost no losses in the 40–59 age group in 2020 but substantial excess mortality in the same group in 2021 (Fig. 2).

Despite the shift towards a greater contribution of excess mortality from younger age groups in 2021, increased mortality among those aged 60+ remained the most important contributor to LE losses compared with pre-pandemic levels (Table 1). LE dropped in 28 of the 29 countries analysed from 2019 to 2021, with only Norway exceeding the 2019 levels. Excess mortality in ages 60+ was the main or sole contributor to these losses in 19 of 28 countries (Austria, Bulgaria, Chile, the Czech Republic, Germany, Estonia, Spain, France, England and Wales, Greece, Croatia, Hungary, Italy, Lithuania, the Netherlands, Poland, Portugal, Slovenia and Slovakia, $H_0: \mu \leq 50\%$ contribution of ages 60+ to LE change; $P < 0.05$), with the United States being the prominent exception. In 2020, the LE losses of every country witnessing significant losses were explained primarily or solely by mortality increases in ages 60+. The LE changes in 2021 were, however, sometimes driven by mortality dynamics below age 60. France stands out as the only country that suffered significant LE losses since 2019 without increased mortality among those under 60 (Table 1 and Fig. 2).

### Sex differences in LE changes

Recent trends of a decreasing gap in LE between females and males[20] were disrupted by the pandemic. Consistent with previous research, females showed higher LE in the 29 countries in our analysis. The magnitude of the gap in 2021 varied from 3.17 years in Norway (95% CI, 2.95 to 3.37; $H_0: \mu \leq 0$; $P < 0.001$) to more than 9.65 years in Lithuania (95% CI, 9.20 to 9.90; $P < 0.001$). However, our results show that the female advantage in LE significantly ($H_0: \mu \leq 0$; $P < 0.05$) increased in 16 of the 29 countries during the pandemic, thereby widening the sex gap (Fig. 3). This finding indicates that in most countries, males were more affected by excess deaths. The biggest increase in the sex gap was observed in the United States, where the gap increased by almost a year from 5.72

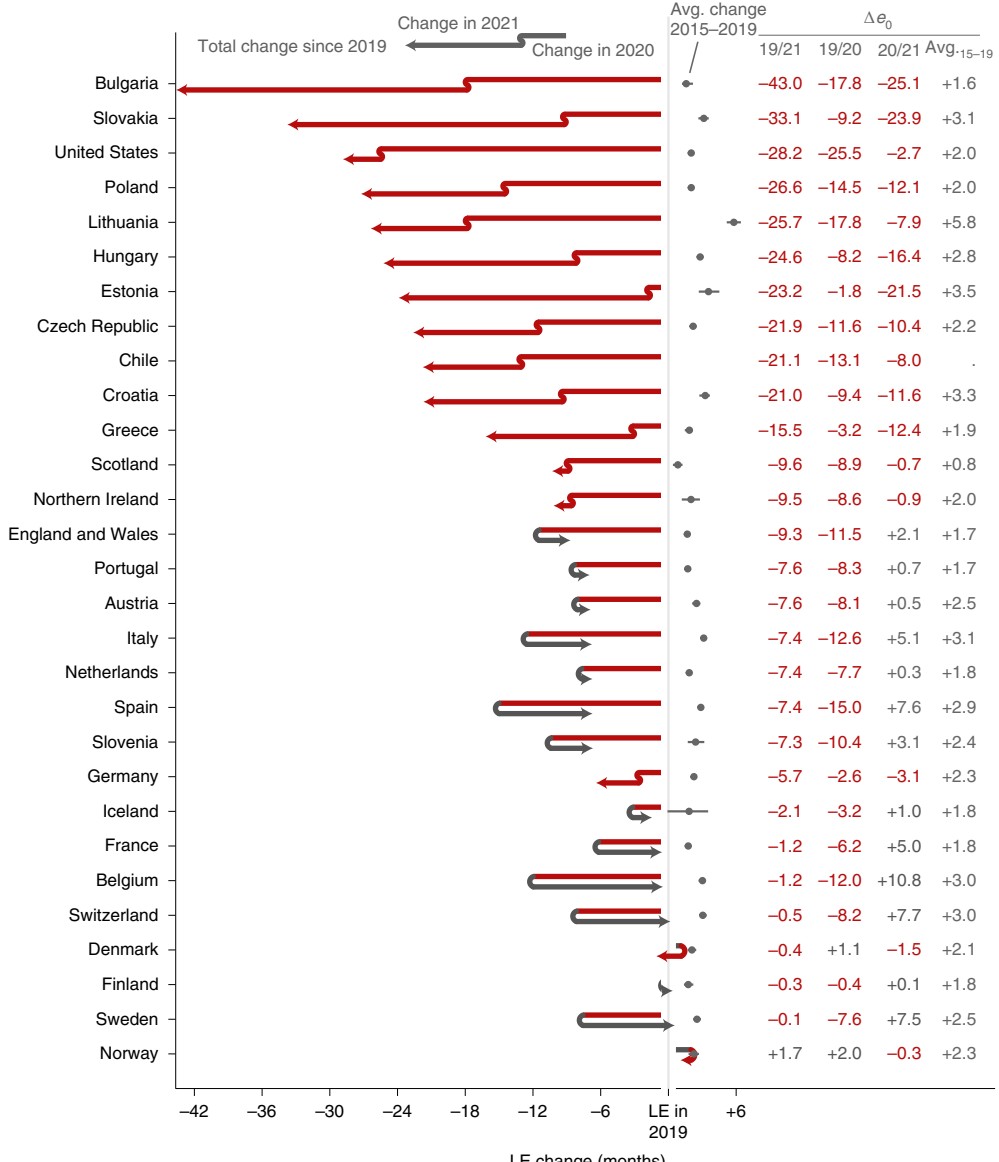

**Fig. 1 | LE changes in 2019–2020 and 2020–2021 across countries.** The countries are ordered by increasing cumulative LE losses since 2019. The two line segments indicate the annual changes in LE in 2020 and 2021. Red segments to the left indicate an LE drop, while grey arrows to the right indicate a rise in LE. The position of the arrowhead indicates the total change in LE from 2019 through 2021. The grey dots and lines indicate the average annual LE changes over the years 2015 through 2019 along with 95% CIs. $\Delta e_0$ marks the change in period LE over the designated period.

to 6.69 years (+0.97 years; 95% CI, 0.90 to 1.04; $H_0$: $\mu \leq 0$; $P < 0.001$). The narrowing sex gaps observed in six countries were not significant. See Extended Data Figs. 4–7 and Supplementary Tables 1 and 2 for further sex-specific results on LE levels and changes.

### LE deficit contributions by cause of death and age

Officially registered COVID-19 deaths explained most of the LE deficit in the year 2021 across Europe, the United States and Chile (Fig. 4). The Netherlands is the single exception where causes other than COVID-19 explained more than half (51.7%) of the 2021 LE deficit. Conversely, France, Slovenia, and England and Wales stand out as three countries where mortality due to non-COVID deaths was lower than expected in 2021. In the majority of countries, the age group 60–79 contributed the most to the LE deficit in 2021. The exceptions were Scotland and Germany, the former having the largest contributions among ages 40–59 (0.45 of the 0.86 years of total LE deficit), the latter among ages 80+ (0.43 of the 0.86 years of total LE deficit). Note that

COVID-19-related deaths are counted differently across countries and that some cross-country differences are explained by different reporting conventions as outlined in the Discussion. As CIs do not account for these biases, we chose to not report them for this section.

### LE deficit by vaccination uptake

Higher vaccination uptake by October 2021 was associated with smaller LE deficits in quarter 4 of 2021 across countries among ages 60+ and <60 ($r_{60+} = -0.86$; two-tailed $P < 0.001$; 95% CI, −0.94 to −0.69; $r_{<60} = -0.74$; two-tailed $P < 0.001$; 95% CI, −0.88 to −0.46; Fig. 5). Eastern Europe, especially Bulgaria, had lower vaccination uptake and showed bigger deficits in LE, while the opposite was true for most central and western European countries.

The direction of this association was the same when comparing the contributions of the age groups <60 and 60+ to the LE deficit in 2021, albeit with variation in the strength of the association. Vaccination uptake for people 60+ showed a stronger association with LE deficits.

**Table 1 | Months of LE changes and deficits (labelled ES) since the start of the pandemic attributed to age-specific mortality changes (labelled AT)**

| | Net LE difference from 2019 to 2021 | | | LE changes in 2020 | | | LE changes in 2021 | | | LE deficit in 2021 | | |
|---|---|---|---|---|---|---|---|---|---|---|---|---|
| | AT¹ | ES² | 95% CI | AT | ES | 95% CI | AT | ES | 95% CI | AT | ES | 95% CI |
| Austria | ↓60+ | −7.6 | (−8.8, −6.4) | ↓60+ | −8.1 | (−9.4, −6.9) | ↑**60+** | +0.5 | (−1.1, +1.7) | ↓60+ | −12.0 | (−10.6, −13.3) |
| Belgium | ↓**60+** | −1.2 | (−2.2, +0.1) | ↓60+ | −12.0 | (−13.0, −10.8) | ↑60+ | +10.8 | (+9.7, +11.9) | ↓60+ | −6.8 | (−5.7, −7.9) |
| Bulgaria | ↓60+ | −43.0 | (−44.5, −41.4) | ↓60+ | −17.8 | (−19.8, −16.2) | ↓60+ | −25.1 | (−26.6, −23.4) | ↓60+ | −46.2 | (−44.5, −48.0) |
| Switzerland | ↓**<60** | −0.5 | (−1.7, +0.6) | ↓60+ | −8.2 | (−9.5, −6.7) | ↑60+ | +7.7 | (+6.4, +8.8) | ↓60+ | −6.2 | (−4.9, −7.4) |
| Chile | ↓60+ | −21.1 | (−22.3, −19.9) | ↓60+ | −13.1 | (−14.2, −11.7) | ↓<60 | −8.0 | (−9.0, −7.0) | ↓60+ | −24.4 | (−23.3, −25.6) |
| Czech Republic | ↓60+ | −21.9 | (−22.8, −20.8) | ↓60+ | −11.6 | (−12.6, −10.5) | ↓60+ | −10.4 | (−11.5, −9.4) | ↓60+ | −26.0 | (−24.9, −27.0) |
| Germany | ↓60+ | −5.7 | (−6.2, −5.3) | ↓**60+** | −2.6 | (−3.1, −2.2) | ↓60+ | −3.1 | (−3.5, −2.7) | ↓60+ | −10.2 | (−9.8, −10.6) |
| Denmark | ↓**60+** | −0.4 | (−1.9, +1.4) | ↑60+ | +1.1 | (−0.3, +2.8) | ↓**60+** | −1.5 | (−3.3, +0.1) | ↓60+ | −3.9 | (−2.2, −5.6) |
| Estonia | ↓60+ | −23.2 | (−26.9, −19.3) | ↓60+ | −1.8 | (−6.5, +3.0) | ↓60+ | −21.5 | (−25.1, −17.6) | ↓60+ | −29.3 | (−25.6, −32.7) |
| Spain | ↓60+ | −7.4 | (−8.0, −6.7) | ↓60+ | −15.0 | (−15.5, −14.4) | ↑**60+** | +7.6 | (+7.1, +8.1) | ↓60+ | −13.3 | (−12.8, −14.0) |
| Finland | ↓**60+** | −0.3 | (−2.0, +1.7) | ↓60+ | −0.4 | (−1.9, +1.4) | ↑**<60** | +0.1 | (−1.5, +1.8) | ↓**60+** | −3.4 | (−1.2, −5.1) |
| France | ↓**60+** | −1.2 | (−1.7, −0.6) | ↓**60+** | −6.2 | (−6.7, −5.6) | ↑60+ | +5.0 | (+4.4, +5.6) | ↓**60+** | −4.5 | (−3.7, −5.1) |
| England and Wales | ↓60+ | −9.3 | (−9.8, −8.9) | ↓60+ | −11.5 | (−12.1, −11.0) | ↑**60+** | +2.1 | (+1.6, +2.7) | ↓60+ | −12.7 | (−12.1, −13.3) |
| Northern Ireland | ↓60+ | −9.5 | (−12.9, −7.0) | ↓**60+** | −8.6 | (−11.4, −5.4) | ↓**<60** | −0.9 | (−3.4, +2.5) | ↓60+ | −13.0 | (−9.4, −15.8) |
| Scotland | ↓60+ | −9.6 | (−11.7, −7.9) | ↓60+ | −8.9 | (−10.7, −7.3) | ↓**<60** | −0.7 | (−2.6, +0.8) | ↓<60 | −10.2 | (−8.5, −11.5) |
| Greece | ↓60+ | −15.5 | (−16.8, −14.2) | ↓**60+** | −3.2 | (−4.3, −1.8) | ↓60+ | −12.4 | (−13.8, −11.0) | ↓60+ | −18.4 | (−17.2, −19.7) |
| Croatia | ↓60+ | −21.0 | (−23.0, −19.5) | ↓60+ | −9.4 | (−11.7, −7.6) | ↓60+ | −11.6 | (−13.3, −9.7) | ↓60+ | −27.0 | (−25.0, −28.9) |
| Hungary | ↓60+ | −24.6 | (−26.0, −23.2) | ↓60+ | −8.2 | (−9.3, −7.0) | ↓60+ | −16.4 | (−17.6, −15.3) | ↓60+ | −29.5 | (−28.3, −30.9) |
| Iceland | ↓**<60** | −2.1 | (−10.5, +5.6) | ↓**<60** | −3.2 | (−10.6, +4.0) | ↑60+ | +1.0 | (−5.3, +8.5) | ↓**<60** | −4.7 | (+3.1, −14.3) |
| Italy | ↓**<60** | −7.4 | (−7.9, −6.9) | ↓60+ | −12.6 | (−13.0, −12.1) | ↑**60+** | +5.1 | (+4.6, +5.5) | ↓60+ | −13.5 | (−13.0, −14.1) |
| Lithuania | ↓60+ | −25.7 | (−28.3, −23.2) | ↓60+ | −17.8 | (−20.4, −15.2) | ↓60+ | −7.9 | (−10.5, −5.4) | ↓60+ | −37.5 | (−34.9, −39.9) |
| Netherlands | ↓60+ | −7.4 | (−8.4, −6.7) | ↓60+ | −7.7 | (−8.5, −6.8) | ↑**60+** | +0.3 | (−0.7, +1.0) | ↓60+ | −11.0 | (−10.1, −12.0) |
| Norway | ↑**<60** | +1.7 | (+0.4, +3.6) | ↑60+ | +2.0 | (+0.4, +3.9) | ↓**60+** | −0.3 | (−2.0, +1.5) | ↓**60+** | −2.4 | (−0.9, −4.4) |
| Poland | ↓60+ | −26.6 | (−27.4, −25.8) | ↓60+ | −14.5 | (−15.4, −13.8) | ↓60+ | −12.1 | (−12.7, −11.3) | ↓60+ | −30.5 | (−29.8, −31.1) |
| Portugal | ↓60+ | −7.6 | (−9.1, −6.5) | ↓60+ | −8.3 | (−9.4, −7.2) | ↑60+ | +0.7 | (−0.8, +2.0) | ↓60+ | −10.8 | (−9.4, −12.1) |
| Sweden | ↓**60+** | −0.1 | (−1.1, +1.0) | ↓60+ | −7.6 | (−8.6, −6.4) | ↑60+ | +7.5 | (+6.0, +8.6) | ↓60+ | −4.9 | (−3.7, −6.0) |
| Slovenia | ↓60+ | −7.3 | (−10.8, −4.8) | ↓60+ | −10.4 | (−13.6, −7.7) | ↑**60+** | +3.1 | (+0.4, +5.7) | ↓60+ | −8.5 | (−5.5, −11.7) |
| Slovakia | ↓60+ | −33.1 | (−34.8, −31.6) | ↓60+ | −9.2 | (−11.2, −7.6) | ↓60+ | −23.9 | (−25.7, −22.3) | ↓60+ | −39.1 | (−37.1, −40.8) |
| United States | ↓<60 | −28.2 | (−28.7, −27.8) | ↓60+ | −25.5 | (−26.0, −25.1) | ↓**<60** | −2.7 | (−3.1, −2.2) | ↓60+ | −33.0 | (−32.4, −33.5) |

¹Attribution of life expectancy changes to mortality increases among primarily ↓60+, solely ↓**60+**, primarily ↓<60 or solely ↓**<60**, or mortality decreases among primarily ↑60+, solely ↑**60+**, primarily ↑<60 or solely ↑**<60**. ²Central estimate in months. LE deficit is defined as observed minus expected LE had pre-pandemic mortality trends continued.

Prominent outliers indicate a confounded relationship between vaccination uptake and LE losses. For those under age 60, the United States saw a far higher LE deficit than countries with comparable overall vaccination shares, such as the Netherlands, Austria and Switzerland. For ages 60+, Slovakia, Croatia and Hungary stand out as countries with surprisingly high LE deficits given their vaccination uptake. Finer-grained details of the age prioritization of vaccine roll-out and the types of vaccines used may account for some of these differences, as well as correlations between vaccine uptake and compliance with non-pharmaceutical interventions or the overall health care system capacity.

## Comparison with past mortality shocks

To contextualize the severity of the LE losses during the COVID-19 pandemic, we examined historical mortality crises over the past 120 years and qualitatively compared them with LE declines since 2019. As shown in Fig. 6, the first half of the twentieth century witnessed several mortality shocks leading to LE declines across consecutive years, but in most cases these were followed by immediate bounce backs. In the past 40 years, the frequency of mortality crises fell markedly. Supplementary Table 3 shows the total negative or overall change in LE during the period of a mortality crisis and the year at which recovery to pre-crisis LE was reached.

During World War I and the Spanish flu epidemic, all countries for which historical data are available experienced substantial losses in LE—the largest annual declines in LE in the past 120 years. In most countries, LE declined continuously throughout the four-year period of the crises, but the losses were the largest in 1918. The steepest declines in LE during 1914–1918 were seen in Italy (−22.7 years) and France (−16.5 years). Denmark, similarly to patterns during COVID-19, experienced the lowest decline in LE during Spanish flu—only one year. Notably, even after all these substantial losses in LE, the recovery to pre-crisis levels was achieved in one or two years (Supplementary Table 3).

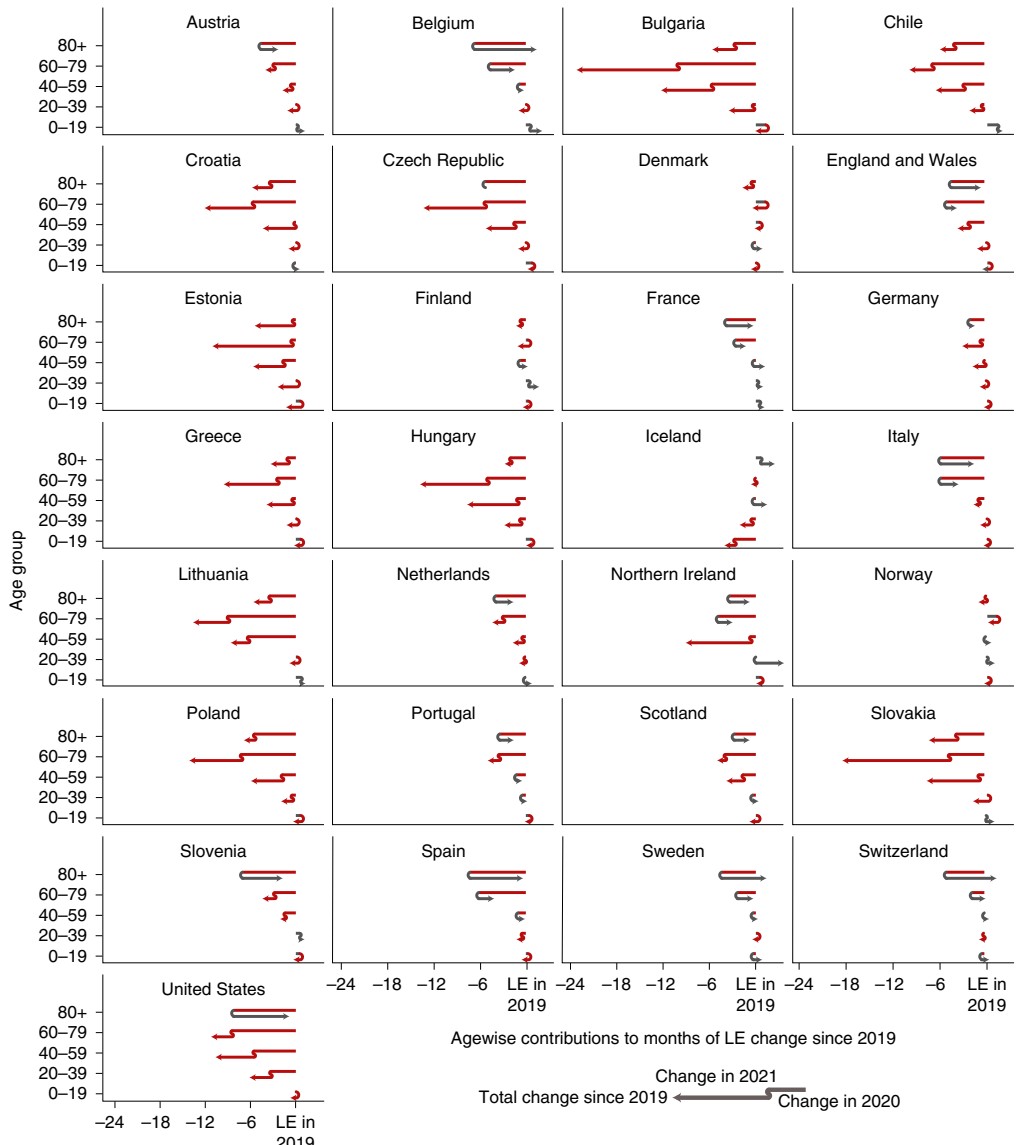

**Fig. 2 | Age contributions to LE changes since 2019 separated for 2020 and 2021.** The position of the arrowhead indicates the contribution of mortality changes in a given age group to the change in LE at birth since 2019. The discontinuity in the arrow indicates those contributions separately for the years 2020 and 2021. Annual contributions can compound or reverse. Red indicates negative contributions, and grey indicates positive contributions. The total LE change from 2019 to 2021 in a given country is the sum of the arrowhead positions across ages.

Mortality patterns during World War II were somewhat different. While most countries suffered some years with substantial LE declines during 1939–1945, the bounce backs occurred during the war or just after it. In the Netherlands, the famine known as Hunger Winter caused LE declines in 1945 of 5.7 years, but LE had already recovered to pre-war (1938) levels by 1946[21].

Numerous flu epidemics occurred over the second half of the twentieth century, but LE during these flu seasons usually declined only very slightly, if at all. Moreover, bounce backs were always immediate, except in cases of wider health crises, such as the recovery from the 2015 influenza season in Scotland and United States, where LE was already stagnating before the epidemic[5]. Overall, the losses during these flu epidemics were substantially smaller than the declines in LE during the COVID-19 pandemic.

The most prominent example of a protracted mortality crisis in the past 50 years is provided by Russia and Eastern Bloc countries. From 1960s onwards, these countries experienced an extended period of continuous stagnation in LE, in which bounce backs to pre-decline

levels were attained only in the twenty-first century[22]. These patterns reflect the deep-rooted structural nature of the mortality crisis in the populations that failed to proceed with the health transition[23]. The most pronounced drop in LE happened in these countries in late 1980s and early 1990s (Fig. 6), which mirrored the previous brief period of success in lowering mortality[24] as a direct result of Gorbachev's anti-alcohol campaign[25]. In contrast to epidemic or war-related shocks, the mortality crisis in the formerly Eastern Bloc countries was structural, and the bounce backs of LE were very slow. The magnitude of LE losses witnessed during COVID-19 in the formerly Eastern Bloc countries are comparable to those observed during the Soviet mortality crisis.

## Discussion

The COVID-19 pandemic led to global increases in mortality and declines in period LE that are without precedent over the past 70 years. The scale of these losses was clear by the end of 2020. By the end of 2021, it was clear that the pandemic had induced a protracted mortality shock in

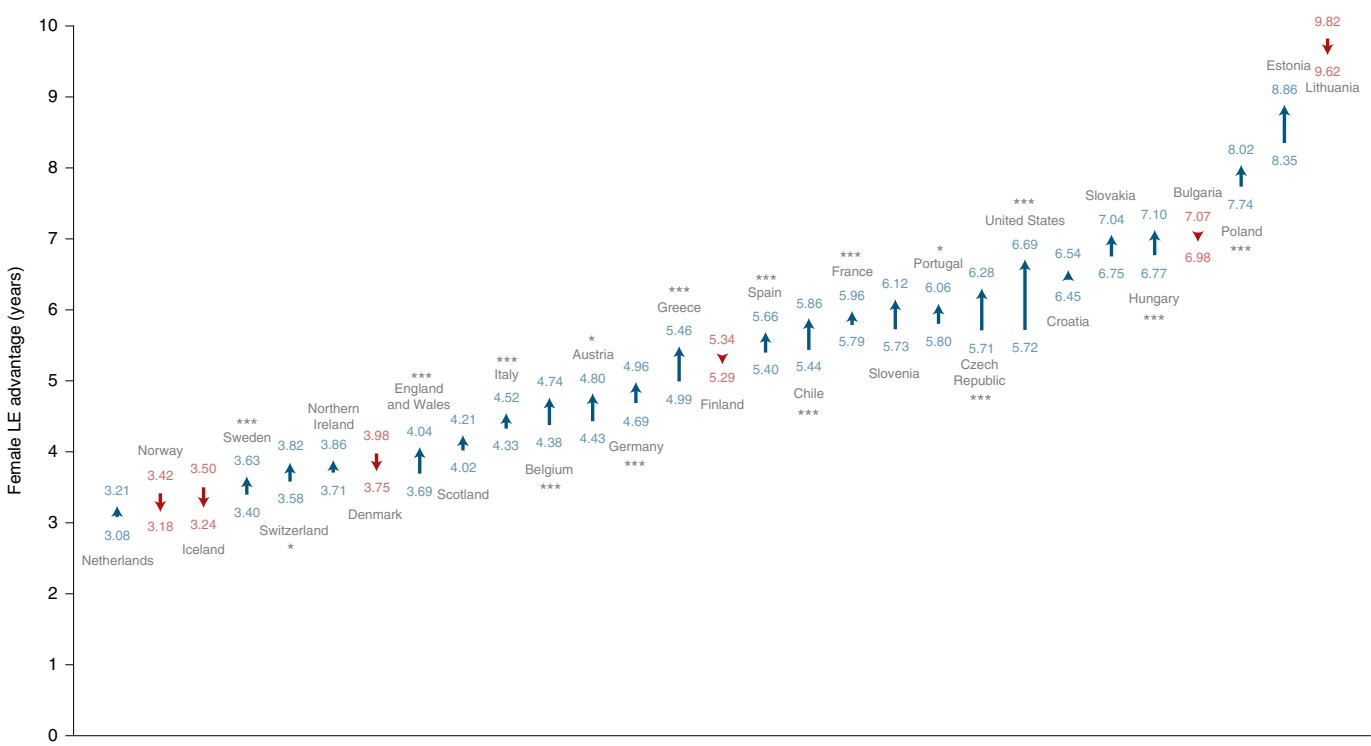

**Fig. 3 | Changes in the female LE advantage from 2019 through 2021.** The blue upwards arrows indicate increases and the red downwards arrows indicate decreases in the female LE advantage. We tested the hypothesis of no change in the sex gap between 2019 and 2021. *$P \le 0.05$; ***$P \le 0.001$.

the United States and many European countries, measured as either compounded LE losses or persistent LE below pre-pandemic levels. Even the best-performing countries were lagging behind their LE projections for 2021 given a continuation of pre-pandemic trends. Nevertheless, in 2021, Sweden, Switzerland, Belgium and France managed to bounce back from substantial LE losses in 2020 to pre-pandemic levels. Demographically, they achieved this by reducing mortality among ages 60+ back to 2019 levels while avoiding the mortality burden shift to younger ages as seen in other countries during 2021.

Denmark, Finland and Norway, while lagging behind their projected LE for 2021, managed to remain at pre-pandemic LE levels throughout 2020 and 2021. Here we may see the combination of campaigns delivering vaccines faster to more people than the European Union average, effective non-pharmaceutical public health interventions and high baseline capacities of the health care systems.

It is important to note that even for those countries that approached pre-pandemic levels of LE in 2021, substantial damage was already done. This is especially relevant for Sweden, which experienced pre-pandemic levels of LE in 2021 but, unlike its Scandinavian neighbours, suffered a substantial LE loss in 2020. While period mortality and consequentially LE can revert back to normal levels, the years of life lost during the period of elevated mortality and declining LE cannot be regained. As a measure of the current mortality conditions in a population, a return to pre-pandemic LE levels simply indicates a normalization of the mortality risk.

We observed stark cross-country differences in 2021 LE deficits, with bigger losses in countries with lower pre-pandemic LE. Geographically, this presents as an East–West division in Europe, which is also aligned with differences in vaccination uptake during 2021, with generally lower vaccination rates in eastern Europe than in the West. This pattern raises questions regarding the future of European mortality convergence, a stated goal of the European Union[26].

Eastern European countries had a distinctive pattern of mortality development in the twentieth century. Early on, highly centralized

governance and directive economies contributed to mortality reductions through the implementation of public health measures, and these countries were very successful in the early stages of their epidemiological transition[27]. Yet, in the second part of the century, these countries witnessed mortality stagnation as these centralized channels were less effective at curbing mortality linked to behavioural factors such as smoking and alcohol[28,29]. Recent years have seen rapid catch-up LE convergence between eastern and western Europe, after the periods of LE declines seen in the 1980s and 1990s[23,30]. It remains to be seen whether persistent COVID-19 losses in eastern Europe and diminishing losses in western Europe will create a new East/West divide and divergence in LE in the years to come.

In the United States, the pandemic has accentuated the pre-existing mid-life mortality crisis. This is clear from the strong contribution of increasing mortality below age 60 to LE losses in 2020 and 2021. Because non-COVID mortality also increased in these ages, this may be interpreted as the continuation and worsening of a pre-existing mortality crisis among working-age adults[31]. In 2020, the largest share of non-COVID excess deaths in US males was from external causes (primarily due to drug overdoses and homicides), nearly 80% of which occurred at working ages[32]. Preliminary data show continued increases in deaths due to drug overdoses in 2021[33]. However, part of the effect may be due to the under-registration of COVID-19 deaths among the working-age population. Differences in vaccine uptake by age may also have contributed to the shift to younger mortality in the United States. By 1 July, when vaccines were already available in the United States, only 66.9% of 50- to 64-year-olds were fully vaccinated compared with 82.3% of 65- to 74-year-olds (as per the COVID Data Tracker of the Centers for Disease Control and Prevention). This means that older age groups were better protected during the large Delta wave in the United States in the summer/autumn of 2021 than during previous waves. Pre-pandemic differences in underlying conditions such as obesity and diabetes may also have contributed to an increased mortality burden in working-age

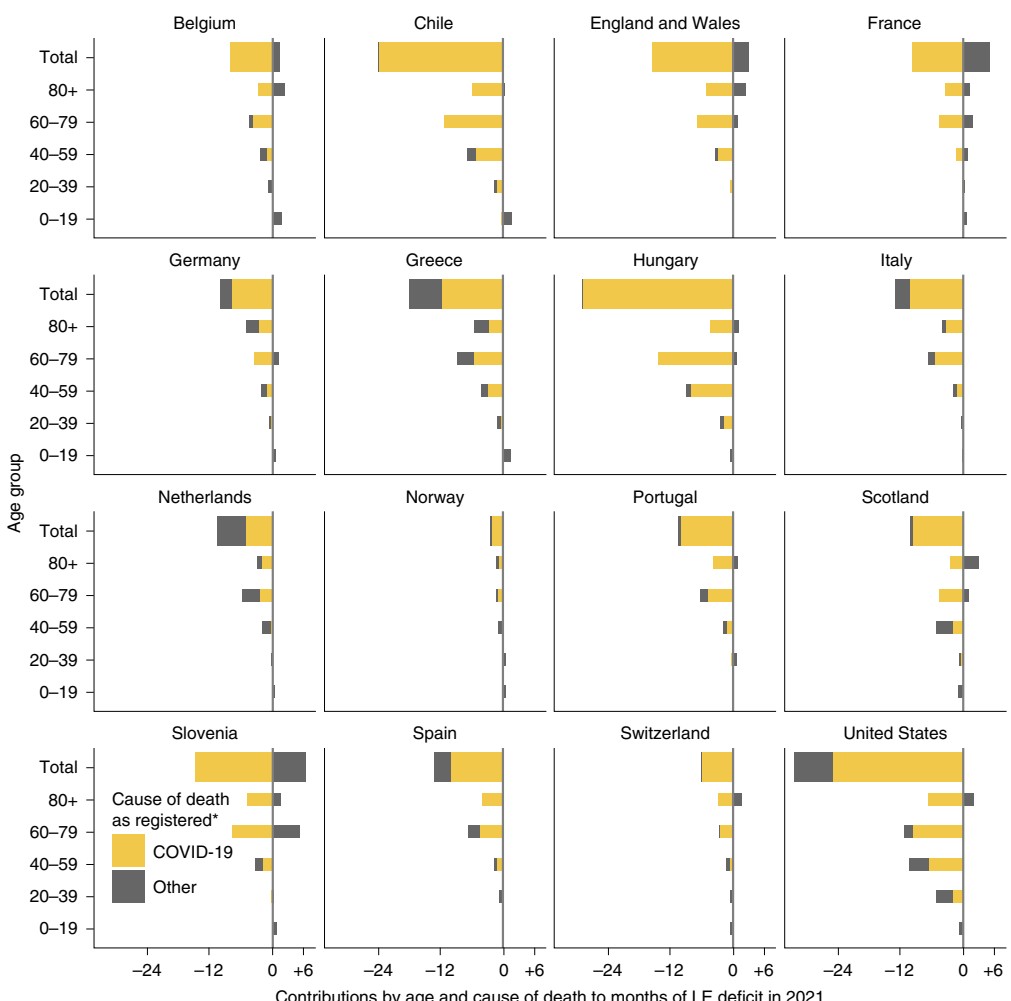

**Fig. 4 | LE deficit in 2021 decomposed into contributions by age and cause of death.** LE deficit is defined as observed minus expected LE had pre-pandemic mortality trends continued.

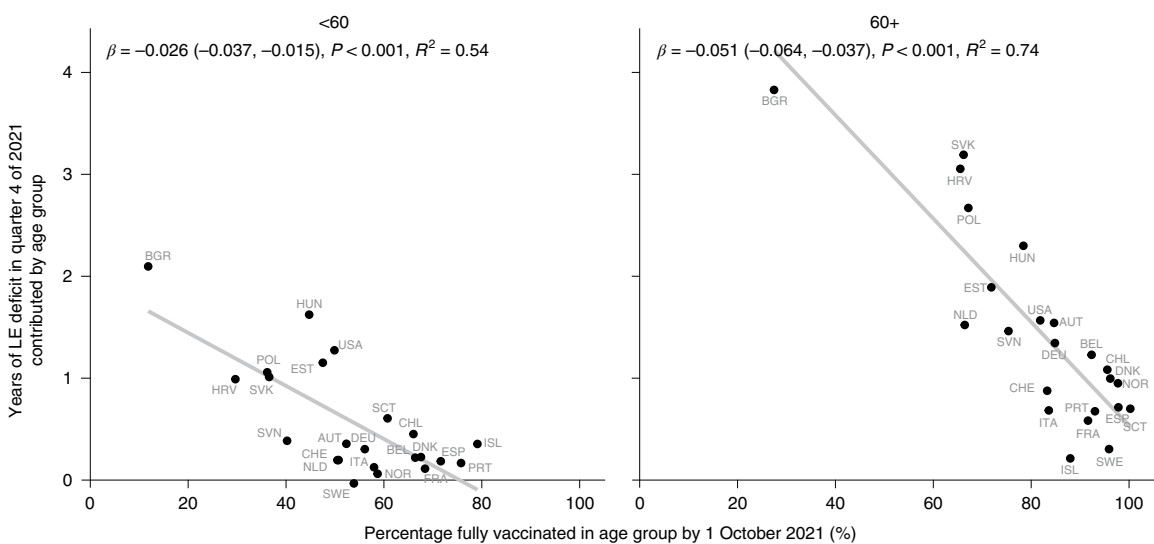

**Fig. 5 | Years of LE deficit in October through December 2021 contributed by ages <60 and 60+ against percentage of population twice vaccinated by 1 October in the respective age groups.** LE deficit is defined as the counterfactual LE from a Lee–Carter mortality forecast based on death rates for the fourth quarter of the years 2015 to 2019 minus observed LE. The points are labelled with ISO three-letter country codes.

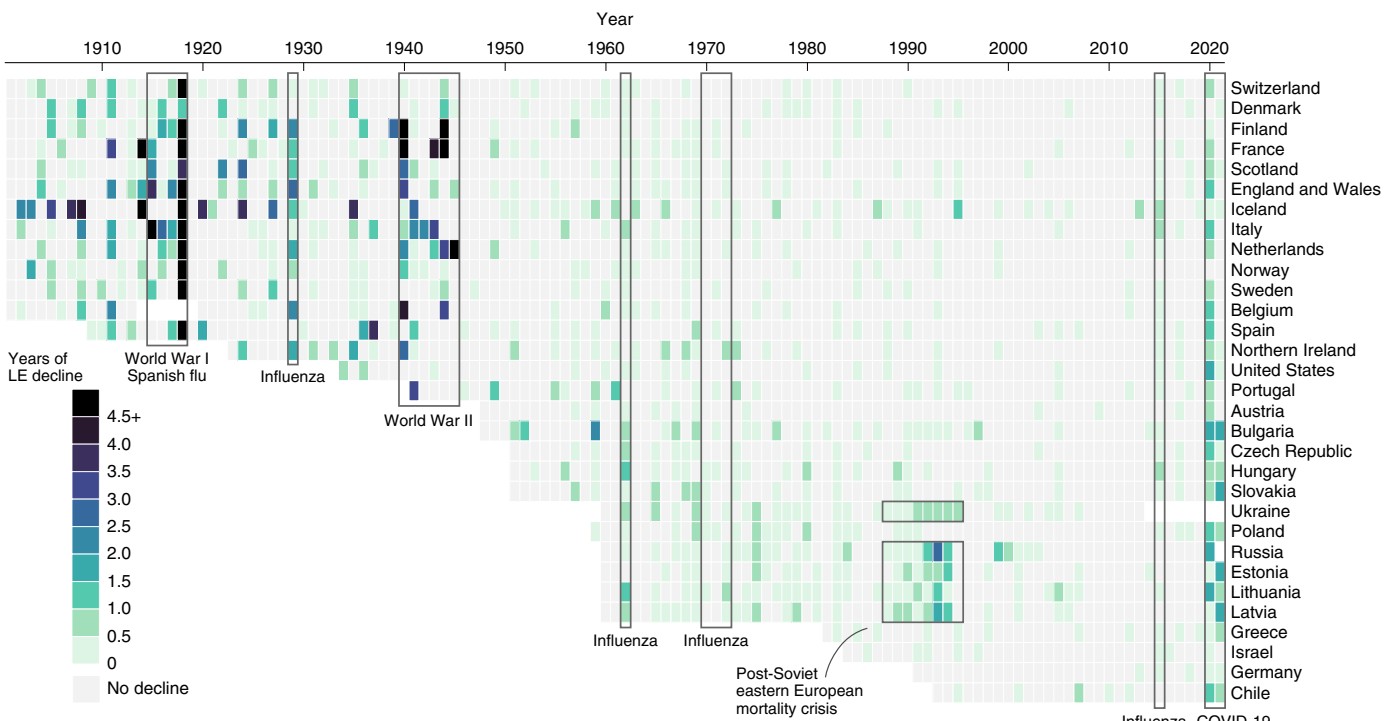

**Fig. 6 | Annual LE declines since 1900.** Period shocks to LE due to wars and epidemics show up as green vertical bands across countries, momentarily disrupting the dominant trend of expectancy improvements, shown in grey.

US adults compared with their European counterparts[34]. Regarding global comparisons, the evidence for co-morbidity prevalence as an important predictor of cross-country COVID-19 mortality differences is still weak[35].

In 2021, the pandemic's death toll shifted to younger ages. While most countries reduced mortality at older ages, countries that bounced back were able to avoid the mortality increases in those under 60 that accounted for a higher share of losses in 2021. Whether this shift towards the young reflects differences in vaccine protection, behavioural responses or deaths from indirect causes remains to be understood. The inconsistent registration of deaths due to COVID-19 across countries[36] complicates any cause-of-death attribution analysis, including ours. We found lower-than-expected mortality due to non-COVID-19 causes in 2021 in Belgium, England and Wales, France, and Slovenia (Fig. 4). Whether these results are an artefact of an overly broad definition of COVID-19-related deaths or point towards a genuine decline in non-COVID mortality (due to, for example, mortality displacement or the lack of flu deaths) is unclear at this point. For France, overcounting COVID-19 deaths seems unlikely, as the underlying data on COVID-19 death counts used here originate from Santé Publique France, which uses a very strict definition of "death due to COVID-19"[36].

COVID-19 was the largest contributor to the 2021 LE deficit in all analysed countries except the Netherlands. This is despite different reporting standards and thus provides additional strong evidence for the direct effect of COVID-19 on increases in mortality. The result for the Netherlands may be spurious, as Karlinsky and Kobak[37] found indirect evidence of substantial under-reporting of deaths due to COVID-19 in the Netherlands during the fall/winter wave in 2021 when comparing excess deaths with COVID-19 deaths (we retrieved updated results for 2021 from the authors' GitHub repository at github.com/dkobak/excess-mortality on 15 February 2021). However, because factors other than undercounting can influence the relationship between COVID-19 deaths and excess deaths, more detailed cause-of-death information

is needed to assess the impact of non-COVID-19-related mortality on 2021 LE deficits.

As the calculation of LE losses requires timely data on population and death counts by age, we are limited in our analysis to those countries with a reliable vital statistics registration system. Consequently, our international analysis based on 29 high- and middle-income countries may give a skewed impression of the global impact of COVID-19 on LE. Indirect LE estimates for 2020–2021 based on excess deaths[18] indicate substantial losses across South America, which match or exceed the losses we estimated for eastern Europe. India and select countries in the Middle East probably had losses on par with the United States, whereas Russia and Mexico suffered LE losses in excess of the 42 months we estimated for Bulgaria. Little is known about the African continent, due to a lack of reliable death registration, and China, due to restricted access to the data. Putting our results in context with these indirect estimates, while the LE losses in central, western and northern Europe over the first two years of the pandemic were drastic given the long trend of declining mortality, they were probably low when compared across the globe.

In 2021, we saw divergence in the impact of the pandemic on population health. While some countries saw bounce backs from stark LE losses thanks to pharmaceutical and non-pharmaceutical public health interventions, others saw sustained and substantial LE deficits. Human populations faced multiple mortality crises during the twentieth century, yet LE kept increasing globally in the medium and long term, especially in the second half of the twentieth century. While COVID-19 has been the most severe global mortality shock since World War II, we will have to wait to know whether and how longer-term LE trends are altered by the pandemic. Extrapolating our findings from 2021, it is plausible that countries with ineffective public health responses will see a protracted health crisis induced by the pandemic, with medium-term stalls in LE improvements, while other regions manage a smoother recovery to return to pre-pandemic trends.

## Methods

### LE estimation and decomposition

The calculation of period LE requires data on death counts and population exposures by single years of age. Timely information on death counts across all causes from 2015 through 2021 was sourced from the Short Term Mortality Fluctuations Input Database[19,38], which continuously collects weekly death counts by age and sex from statistical offices. The 38 countries represented in the database have been selected on the basis of the completeness of their death registration and census data. To allow for a reliable estimation of LE changes and their age contributions, we further selected the 29 countries for which death counts for 2020 and 2021 were reported across at least ten distinct age groups. These weekly counts were aggregated into annual counts and, due to various age groupings used in the source data, harmonized into single age groups from 0 to 85+ using the penalized composite link model[39,40], a non-parametric disaggregation technique for histograms of count data. Separate tables of harmonized counts were created for males, females and the total population in each country. Our methodology for age disaggregation follows that of Aburto et al.[6] and has been thoroughly validated for accurate LE estimates consistent with those based on unabridged data. Mid-year population estimates by age and sex for the years 2015–2021 were sourced from the United Nations World Population Prospects (WPP)[41] and converted into person-years of exposures, taking into account the varying number of weeks in an ISO year[42] to be consistent with death-count reporting in our raw data. For Scotland and England and Wales, we projected mid-year population for 2020 and 2021 on the basis of data from the respective national statistical offices (NSOs). A sensitivity analysis with alternative population estimates taken from NSOs is provided in the Supplementary Information. To attribute changes in LE to changes in mortality from deaths registered as due to COVID-19, we sourced age- and sex-specific COVID-19 death counts from the COVerAGE-DB database[43]. All data were downloaded on 26 April 2022.

Annual period life tables calculated via standard demographic techniques[44] were used to estimate LE from 2015 to 2021. To quantify the LE loss directly attributable to COVID-19, we further assembled cause-deleted life tables with 'deaths registered as due to COVID-19' and 'other deaths' as possible decrements[45]. Using the Arriaga decomposition technique[46], we attributed annual changes in LE to changes in age-specific all-cause mortality. Additionally, we calculated LE deficits for 2020 and 2021 defined as observed LE minus expected LE based on a continuation of pre-pandemic trends. These expected LEs were derived from Lee–Carter forecasts[47] of age-specific death rates over the years 2015 through 2019. We performed an Arriaga decomposition of the LE deficits in 2021 into age-specific contributions from COVID-19 and non-COVID-19 mortality. CIs around our LE estimates, LE differences, LE deficits, LE decompositions and LE sex differences were derived from 100 Poisson simulations of the harmonized death counts. Where applicable, we calculated the associated $P$ values from a shifted empirical distribution function of the Poisson-bootstrapped estimates under the null.

To analyse the cross-country relationship between LE deficits and vaccination uptake during fall and winter 2021, we calculated life tables and LE deficits for the fourth quarter of the year. To do so, we used the weekly death counts over weeks 40 through 52 and adjusted population exposures.

The analyses are fully reproducible with source code and data archived with Zenodo (https://doi.org/10.5281/zenodo.6861804). We also archived CSV files of the table and figure data (https://doi.org/10.5281/zenodo.6861843) and the harmonized life table data informing all analyses in this paper (https://doi.org/10.5281/zenodo.6861866).

### Estimation of age-specific vaccination uptake

Data on age-specific vaccinations were collected from the COVerAGE-DB[43]. Vaccination uptake was calculated with mid-year population data for 2020 from the WPP[41] and the Human Mortality Database. The vaccination measures were age harmonized using the penalized composite link model. Vaccination counts with missing age information were redistributed according to the observed age distribution of vaccinations in a given country. From the available data, for each country we calculated the share of the population that was fully vaccinated as of 1 October 2021 separately for people below age 60 and above 60. A full vaccination was defined as either two vaccinations or a single vaccination of the vaccine from Johnson & Johnson.

### Population exposure sensitivity analysis

The results presented in the main text rely on population exposures from the WPP, which were issued prior to the COVID-19 pandemic. The COVID-19 pandemic has altered population age structure in some countries, reducing the number of people ages 60 and older. The WPP population projections used in this paper do not account for this effect and thus may overestimate the population in the oldest age groups in 2020 and 2021, resulting in negatively biased death rates and positively biased bounce backs. To test how sensitive our results are to the WPP-based denominators, we performed a sensitivity analysis to compare WPP-based denominators used in the analysis with population estimates (or projections in most cases for 2021) reported by the NSOs of the countries analysed. We then assessed the effects of using these two different sources of population exposures on LE estimates.

As shown in Supplementary Table 4, the overall differences between WPP and NSO mid-year population estimates for 2019, 2020 and 2021 (when available) are generally small and are not consistently positive or negative. Countries that show the largest deviations (such as France) in fact show positive deviations; that is, WPP-based population estimates are smaller than those reported for NSOs for 2020. In age-specific comparisons, we find a high correlation between the WPP- and NSO-based population estimates.

Supplementary Fig. 1 shows estimates of LE using NSO-based denominators compared to WPP-based denominators for 2019, 2020 and 2021 when available. LE estimates using the two different sources for denominators are concordant and largely lie on or close to the $x = y$ line. When the two diverge, the divergence shows NSO-based estimates slightly overestimating LE relative to WPP estimates in 2019. While the direction of the LE changes across both sources is thus the same, the more optimistic LE levels in 2019 imply that NSO-based denominators indicate slightly larger losses in LE between 2019 and 2020, and for the few countries for which 2021 estimates are available, more positive bounce backs in 2021.

### Reporting summary

Further information on research design is available in the Nature Research Reporting Summary linked to this article.

## Data availability

We archived CSV files of the table and figure data (https://doi.org/10.5281/zenodo.6861843) and the harmonized life table data informing all analyses in this paper (https://doi.org/10.5281/zenodo.6861866).

## Code availability

Scripts for the R programming language to download the source data for our analysis and to reproduce the results in this paper were archived with Zenodo (https://doi.org/10.5281/zenodo.6861804).

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

## Acknowledgements

We thank A. Karlinsky for his help in determining the death count late registration quotas via his invaluable work on the World Mortality Database. We acknowledge the following funding: European Union Horizon 2020 research and innovation programme under Marie Sklodowska-Curie grant agreement no. 896821 (J.M.A. and R.K.), ROCKWOOL Foundation's Excess Deaths grant (J.M.A. and I.K.), the Leverhulme Trust Large Centre Grant (J.M.A., L.Z., R.K. and J.B.D.), European Research Council grant no. ERC-2021-CoG-101002587 (MORTAL) (J.B.D.), the University of Oxford John Fell Fund (J.M.A., L.Z., R.K. and J.B.D.) and Estonian Research Council grant no. PSG 669 (H.J.). The funders had no role in study design, data collection and analysis, decision to publish or preparation of the manuscript.

## Author contributions

Conceptualization: J.S., J.M.A., I.K., M.S.K., H.J. and R.K. Data curation: J.S., M.S.K., I.K., L.Z., H.J. and R.K. Formal analysis: J.S., I.K., M.S.K., L.Z., H.J. and R.K. Investigation: J.S., J.M.A., I.K., M.S.K., L.Z., H.J, J.B.D. and R.K. Methodology: J.S., J.M.A., M.S.K., L.Z. and R.K. Project administration: J.S. Software: J.S., M.S.K., I.K. and H.J. Supervision: J.S., J.M.A. and R.K. Validation: J.S., J.M.A., I.K., M.S.K., L.Z., H.J., J.B.D. and R.K. Visualization: J.S., I.K. and H.J. Writing—original draft: J.S., J.M.A., I.K., M.S.K., L.Z., H.J., J.B.D. and R.K. Writing—review and editing: J.S., J.M.A., I.K., M.S.K., L.Z., H.J., J.B.D. and R.K.

## Funding

## Competing interests

The authors declare no competing interests.

## Additional information

**Extended data** is available for this paper at https://doi.org/10.1038/s41562-022-01450-3.

**Correspondence and requests for materials** should be addressed to Jonas Schöley, José Manuel Aburto or Ridhi Kashyap.

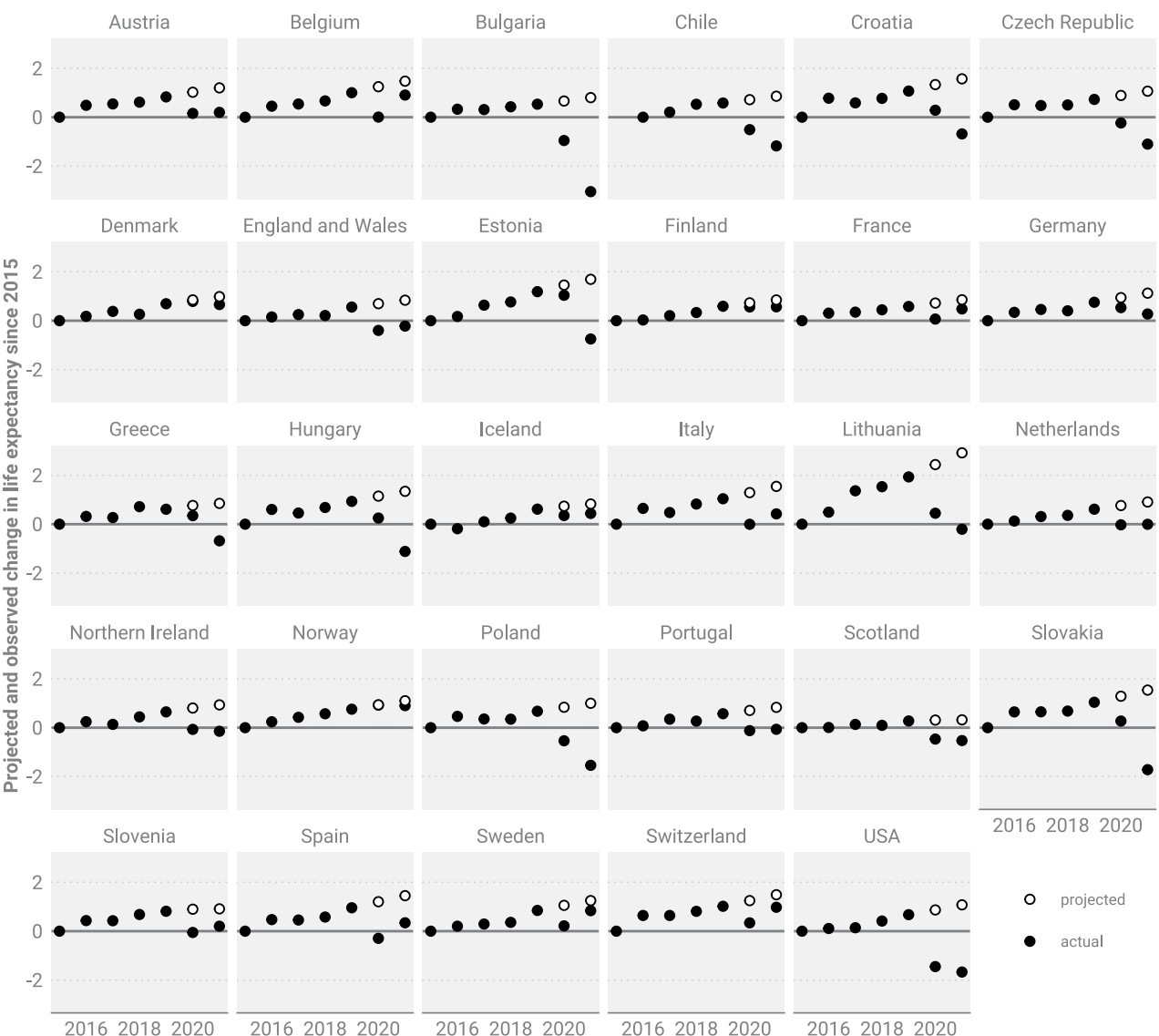

**Extended Data Fig. 1 | Actual and forecast total population life expectancy change since 2015.** LE forecasts are based on the Lee-Carter model based upon the assumption that pre-pandemic mortality trends would have continued into 2020 and 2021.

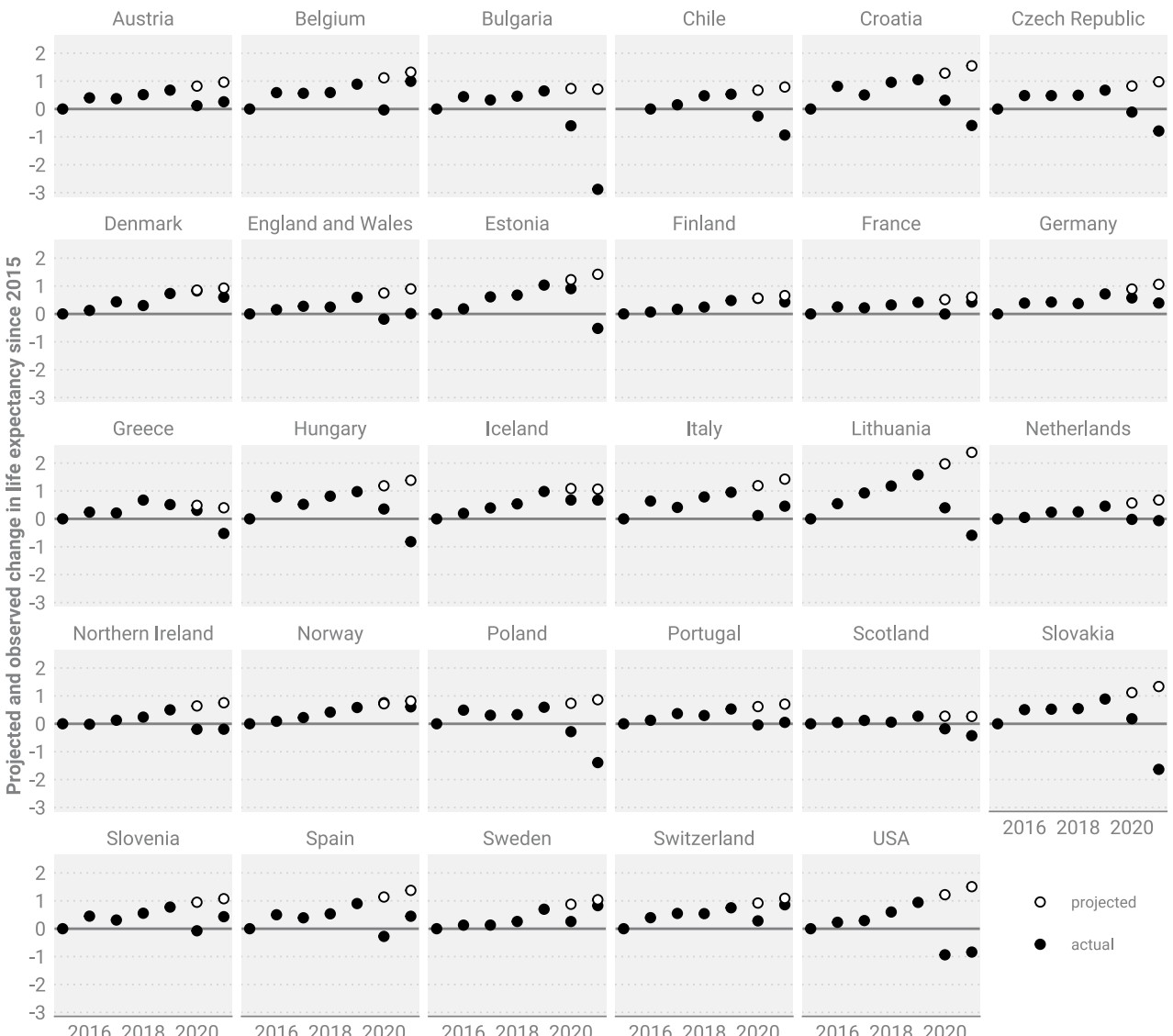

**Extended Data Fig. 2 | Actual and forecast female life expectancy change since 2015.** LE forecasts are based on the Lee-Carter model based upon the assumption that pre-pandemic mortality trends would have continued into 2020 and 2021.

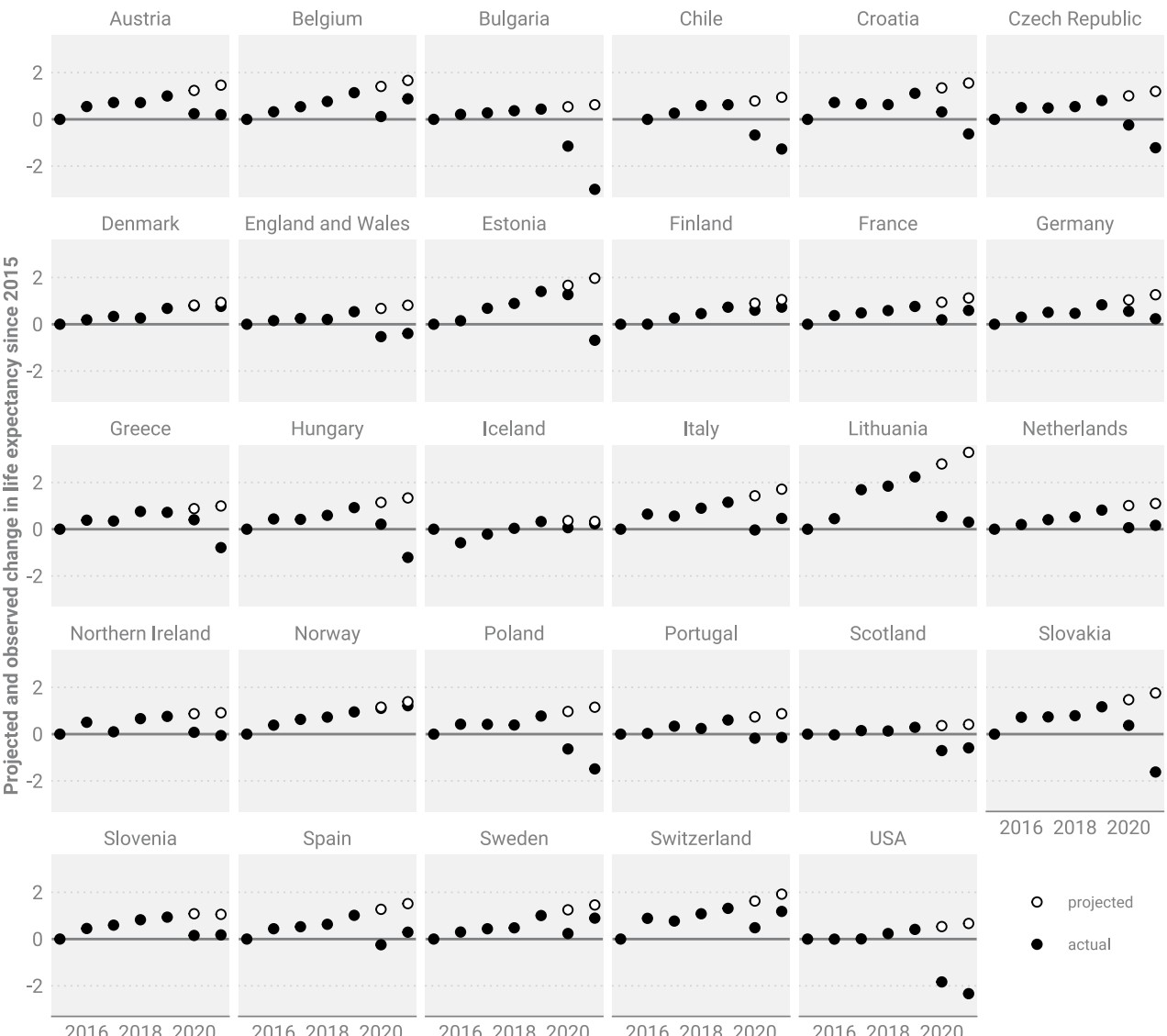

**Extended Data Fig. 3 | Actual and forecast male life expectancy change since 2015.** LE forecasts are based on the Lee-Carter model based upon the assumption that pre-pandemic mortality trends would have continued into 2020 and 2021.

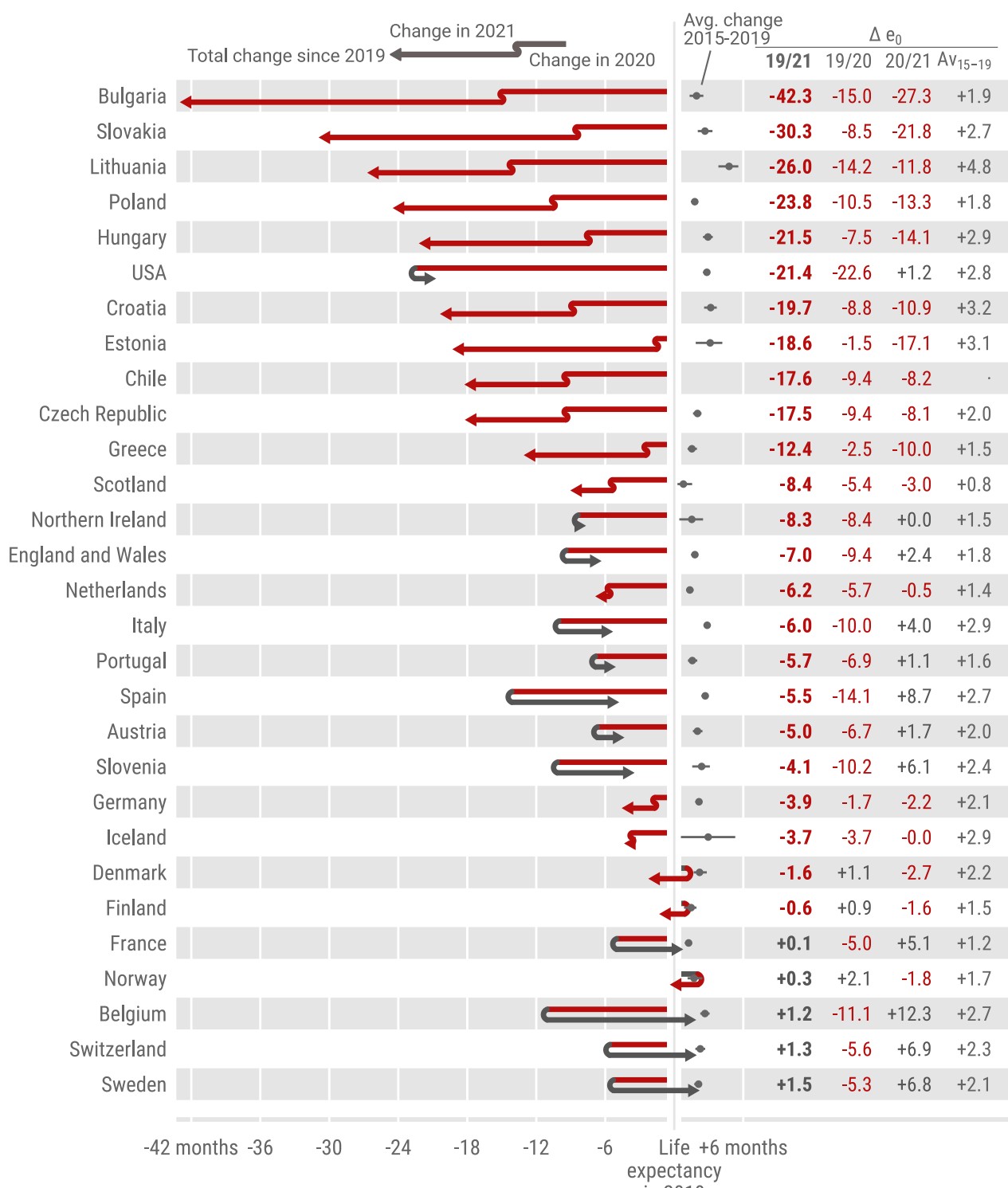

**Extended Data Fig. 4 | Female life expectancy changes 2019-20 and 2020-21 across countries.** The countries are ordered by increasing cumulative life expectancy losses since 2019. The two line segments indicate the annual changes in life expectancy in 2020 and 2021 respectively. Red segments to the left indicate a life expectancy drop while gray arrows to the right indicate a rise in life expectancy. The position of the arrowhead indicates the total change in life expectancy from 2019 through 2021. Grey dots and lines indicate the average annual LE changes over the years 2015 through 2019 along with 95% confidence intervals.

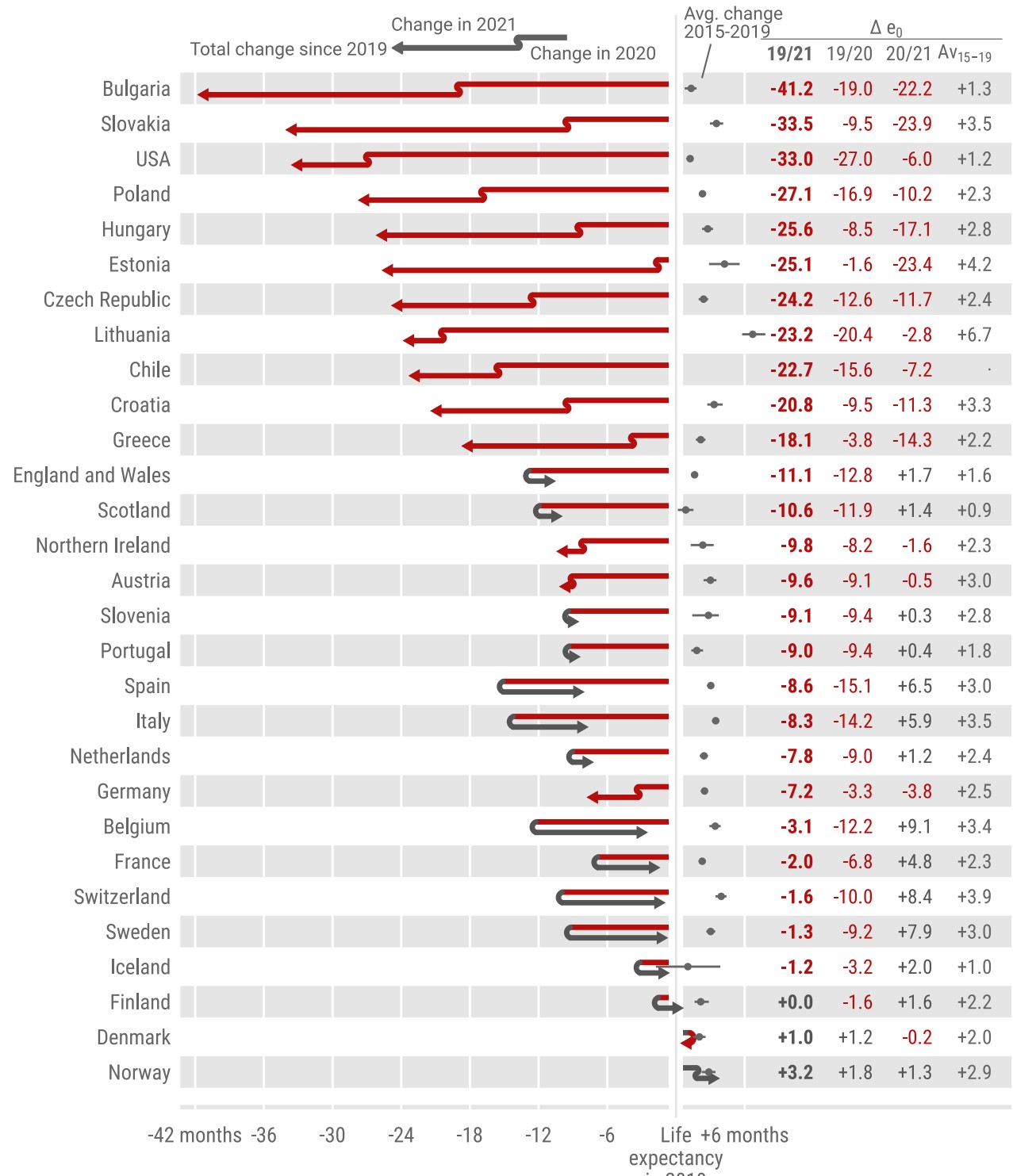

| | | Δ $e_0$ | | |
|---|---|---|---|---|
| | **19/21** | 19/20 | 20/21 | Av$_{15-19}$ |
| Bulgaria | **-41.2** | -19.0 | -22.2 | +1.3 |
| Slovakia | **-33.5** | -9.5 | -23.9 | +3.5 |
| USA | **-33.0** | -27.0 | -6.0 | +1.2 |
| Poland | **-27.1** | -16.9 | -10.2 | +2.3 |
| Hungary | **-25.6** | -8.5 | -17.1 | +2.8 |
| Estonia | **-25.1** | -1.6 | -23.4 | +4.2 |
| Czech Republic | **-24.2** | -12.6 | -11.7 | +2.4 |
| Lithuania | **-23.2** | -20.4 | -2.8 | +6.7 |
| Chile | **-22.7** | -15.6 | -7.2 | · |
| Croatia | **-20.8** | -9.5 | -11.3 | +3.3 |
| Greece | **-18.1** | -3.8 | -14.3 | +2.2 |
| England and Wales | **-11.1** | -12.8 | +1.7 | +1.6 |
| Scotland | **-10.6** | -11.9 | +1.4 | +0.9 |
| Northern Ireland | **-9.8** | -8.2 | -1.6 | +2.3 |
| Austria | **-9.6** | -9.1 | -0.5 | +3.0 |
| Slovenia | **-9.1** | -9.4 | +0.3 | +2.8 |
| Portugal | **-9.0** | -9.4 | +0.4 | +1.8 |
| Spain | **-8.6** | -15.1 | +6.5 | +3.0 |
| Italy | **-8.3** | -14.2 | +5.9 | +3.5 |
| Netherlands | **-7.8** | -9.0 | +1.2 | +2.4 |
| Germany | **-7.2** | -3.3 | -3.8 | +2.5 |
| Belgium | **-3.1** | -12.2 | +9.1 | +3.4 |
| France | **-2.0** | -6.8 | +4.8 | +2.3 |
| Switzerland | **-1.6** | -10.0 | +8.4 | +3.9 |
| Sweden | **-1.3** | -9.2 | +7.9 | +3.0 |
| Iceland | **-1.2** | -3.2 | +2.0 | +1.0 |
| Finland | **+0.0** | -1.6 | +1.6 | +2.2 |
| Denmark | **+1.0** | +1.2 | -0.2 | +2.0 |
| Norway | **+3.2** | +1.8 | +1.3 | +2.9 |

**Extended Data Fig. 5 | Male life expectancy changes 2019-20 and 2020-21 across countries.** The countries are ordered by increasing cumulative life expectancy losses since 2019. The two line segments indicate the annual changes in life expectancy in 2020 and 2021 respectively. Red segments to the left indicate a life expectancy drop while gray arrows to the right indicate a rise in life expectancy. The position of the arrowhead indicates the total change in life expectancy from 2019 through 2021. Grey dots and lines indicate the average annual LE changes over the years 2015 through 2019 along with 95% confidence intervals.

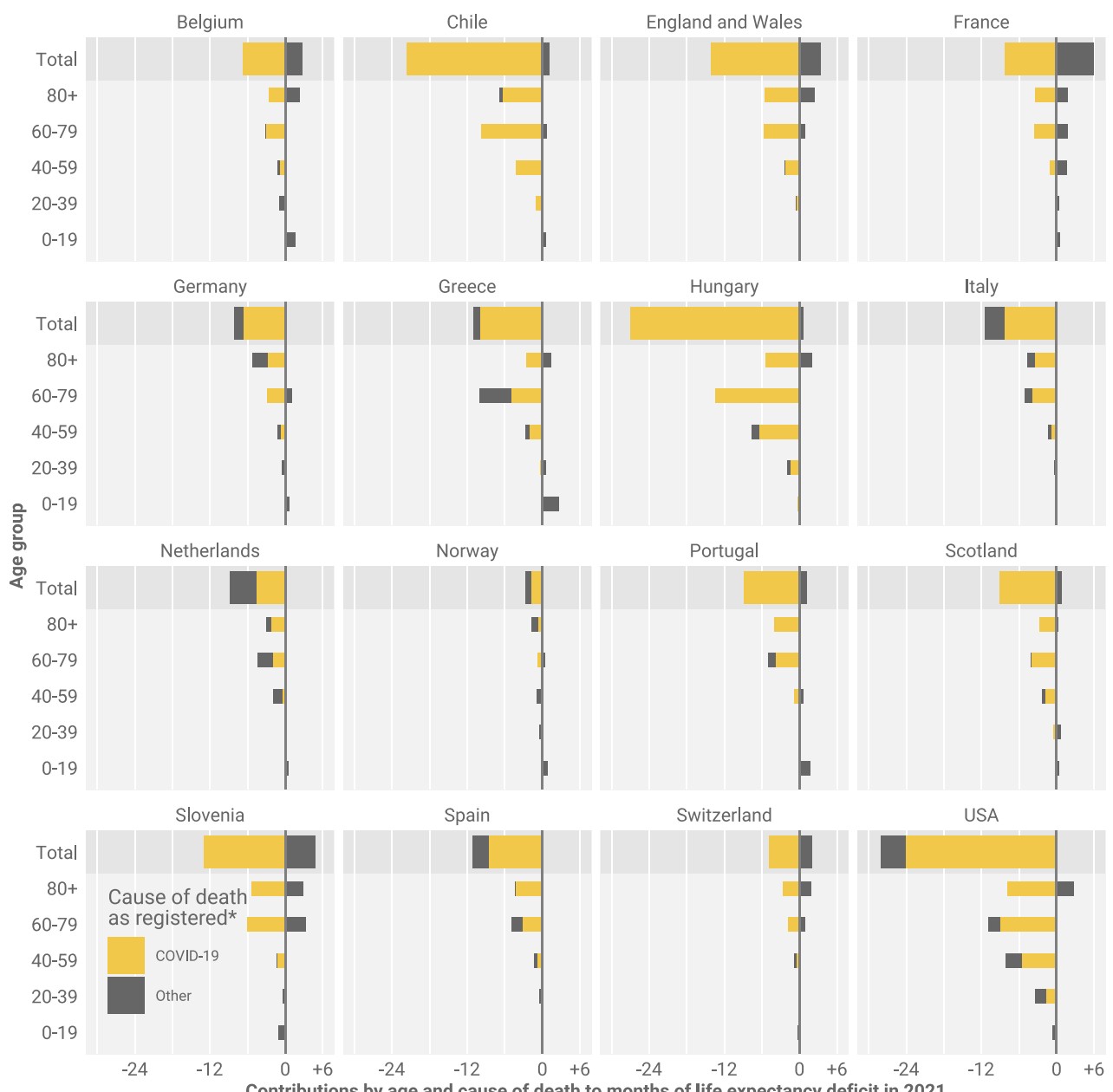

*Note that COVID-19 related deaths are counted differently across countries.
Some of the cross-country differences will be explained by different reporting conventions.

**Extended Data Fig. 6 | Female life expectancy deficit in 2021 decomposed into contributions by age and cause of death.** LE deficit is defined as observed minus expected life expectancy had pre-pandemic mortality trends continued.

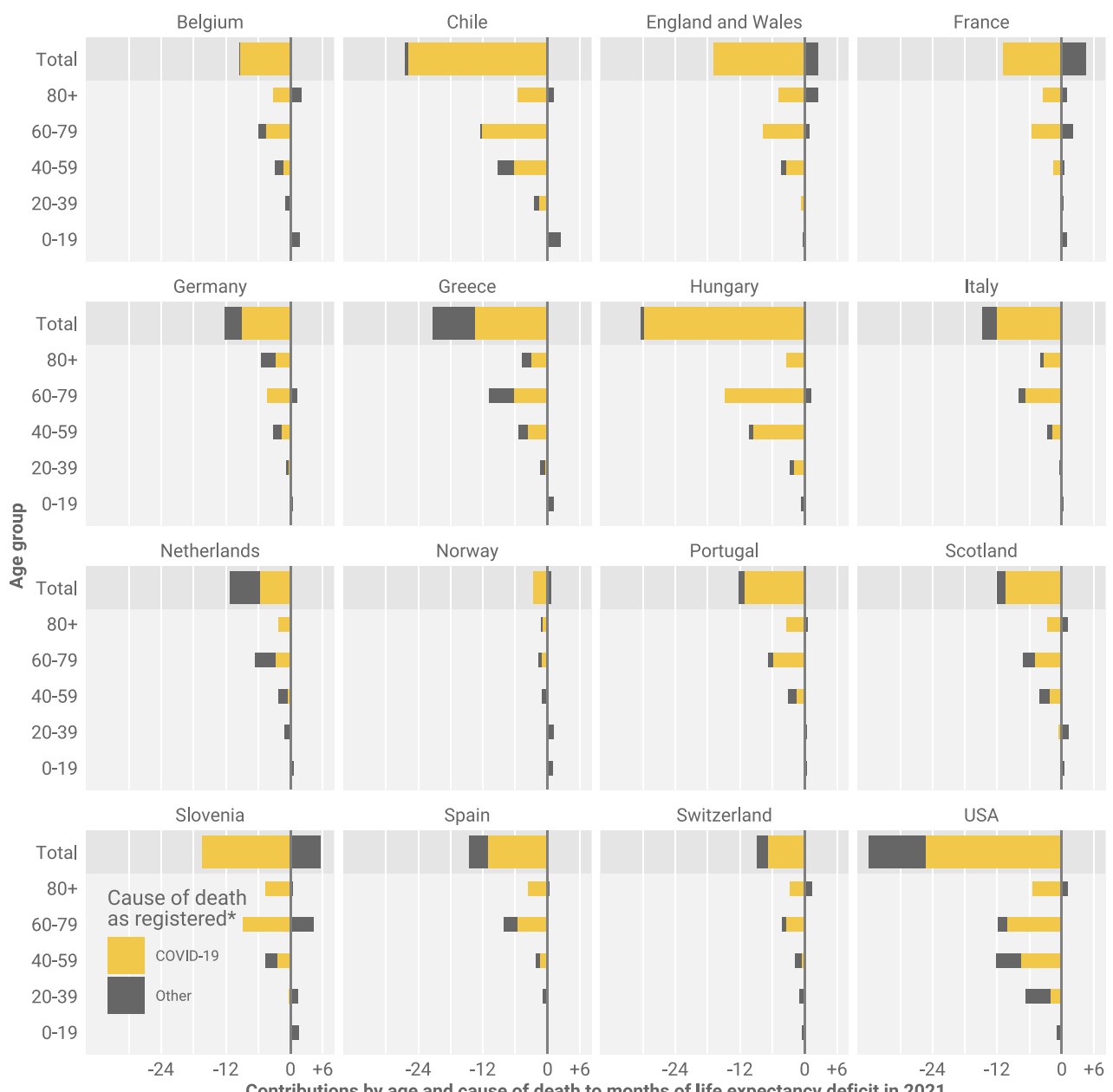

*Note that COVID-19 related deaths are counted differently across countries.
Some of the cross-country differences will be explained by different reporting conventions.

**Extended Data Fig. 7 | Male life expectancy deficit in 2021 decomposed into contributions by age and cause of death.** LE deficit is defined as observed minus expected life expectancy had pre-pandemic mortality trends continued.

José Manuel Aburto
Ridhi Kashyap

# Reporting Summary

## Statistics

For all statistical analyses, confirm that the following items are present in the figure legend, table legend, main text, or Methods section.

| n/a | Confirmed | |
|---|---|---|
| ☒ | ☐ | The exact sample size ($n$) for each experimental group/condition, given as a discrete number and unit of measurement |
| ☐ | ☒ | A statement on whether measurements were taken from distinct samples or whether the same sample was measured repeatedly |
| ☐ | ☒ | The statistical test(s) used AND whether they are one- or two-sided *Only common tests should be described solely by name; describe more complex techniques in the Methods section.* |
| ☐ | ☒ | A description of all covariates tested |
| ☐ | ☒ | A description of any assumptions or corrections, such as tests of normality and adjustment for multiple comparisons |
| ☐ | ☒ | A full description of the statistical parameters including central tendency (e.g. means) or other basic estimates (e.g. regression coefficient) AND variation (e.g. standard deviation) or associated estimates of uncertainty (e.g. confidence intervals) |
| ☐ | ☒ | For null hypothesis testing, the test statistic (e.g. $F$, $t$, $r$) with confidence intervals, effect sizes, degrees of freedom and $P$ value noted *Give P values as exact values whenever suitable.* |
| ☒ | ☐ | For Bayesian analysis, information on the choice of priors and Markov chain Monte Carlo settings |
| ☒ | ☐ | For hierarchical and complex designs, identification of the appropriate level for tests and full reporting of outcomes |
| ☐ | ☒ | Estimates of effect sizes (e.g. Cohen's $d$, Pearson's $r$), indicating how they were calculated |

*Our web collection on statistics for biologists contains articles on many of the points above.*

## Software and code

Policy information about availability of computer code

| | |
|---|---|
| Data collection | Scripts for the R programming language to download the data from public sources are available from Zenodo DOI 10.5281/zenodo.6861804. |
| Data analysis | Scripts for the R programming language to reproduce the data analysis are available from Zenodo DOI 10.5281/zenodo.6861804. |

For manuscripts utilizing custom algorithms or software that are central to the research but not yet described in published literature, software must be made available to editors and reviewers. We strongly encourage code deposition in a community repository (e.g. GitHub). See the Nature Portfolio guidelines for submitting code & software for further information.

## Data

Policy information about availability of data

All manuscripts must include a data availability statement. This statement should provide the following information, where applicable:
- Accession codes, unique identifiers, or web links for publicly available datasets
- A description of any restrictions on data availability
- For clinical datasets or third party data, please ensure that the statement adheres to our policy

We archived CSV files of the table and figure data (DOI: 10.5281/zenodo.6861843), and the harmonized life table data informing all analyses in this paper (DOI: 10.5281/zenodo.6861866).

# Field-specific reporting

Please select the one below that is the best fit for your research. If you are not sure, read the appropriate sections before making your selection.

☐ Life sciences ☒ Behavioural & social sciences ☐ Ecological, evolutionary & environmental sciences

For a reference copy of the document with all sections, see nature.com/documents/nr-reporting-summary-flat.pdf

# Life sciences study design

All studies must disclose on these points even when the disclosure is negative.

| Sample size | *Describe how sample size was determined, detailing any statistical methods used to predetermine sample size OR if no sample-size calculation was performed, describe how sample sizes were chosen and provide a rationale for why these sample sizes are sufficient.* |
|---|---|
| Data exclusions | *Describe any data exclusions. If no data were excluded from the analyses, state so OR if data were excluded, describe the exclusions and the rationale behind them, indicating whether exclusion criteria were pre-established.* |
| Replication | *Describe the measures taken to verify the reproducibility of the experimental findings. If all attempts at replication were successful, confirm this OR if there are any findings that were not replicated or cannot be reproduced, note this and describe why.* |
| Randomization | *Describe how samples/organisms/participants were allocated into experimental groups. If allocation was not random, describe how covariates were controlled OR if this is not relevant to your study, explain why.* |
| Blinding | *Describe whether the investigators were blinded to group allocation during data collection and/or analysis. If blinding was not possible, describe why OR explain why blinding was not relevant to your study.* |

# Behavioural & social sciences study design

All studies must disclose on these points even when the disclosure is negative.

| Study description | Cross-country time-series analysis of population level mortality |
|---|---|
| Research sample | Aggregate data representing the complete population of 29 countries from 2015 through 2021 |
| Sampling strategy | Complete population sample via vital registration systems |
| Data collection | Public aggregate data on demographics, vaccinations, and deaths by HMD/HFD/UN/COVerAGE-DB further harmonized by us. |
| Timing | Data downloaded Apr 26, 2022. |
| Data exclusions | Out of 38 countries we excluded 9 due to insufficient information on ages at death. |
| Non-participation | Participation was based upon data availability in the HMD Short Term Mortality Fluctuations Database. |
| Randomization | N/A due to complete sample. |

# Ecological, evolutionary & environmental sciences study design

All studies must disclose on these points even when the disclosure is negative.

| Study description | *Briefly describe the study. For quantitative data include treatment factors and interactions, design structure (e.g. factorial, nested, hierarchical), nature and number of experimental units and replicates.* |
|---|---|
| Research sample | *Describe the research sample (e.g. a group of tagged Passer domesticus, all Stenocereus thurberi within Organ Pipe Cactus National Monument), and provide a rationale for the sample choice. When relevant, describe the organism taxa, source, sex, age range and any manipulations. State what population the sample is meant to represent when applicable. For studies involving existing datasets, describe the data and its source.* |
| Sampling strategy | *Note the sampling procedure. Describe the statistical methods that were used to predetermine sample size OR if no sample-size calculation was performed, describe how sample sizes were chosen and provide a rationale for why these sample sizes are sufficient.* |
| Data collection | *Describe the data collection procedure, including who recorded the data and how.* |
| Timing and spatial scale | *Indicate the start and stop dates of data collection, noting the frequency and periodicity of sampling and providing a rationale for* |

| Timing and spatial scale | *these choices. If there is a gap between collection periods, state the dates for each sample cohort. Specify the spatial scale from which the data are taken* |
| Data exclusions | *If no data were excluded from the analyses, state so OR if data were excluded, describe the exclusions and the rationale behind them, indicating whether exclusion criteria were pre-established.* |
| Reproducibility | *Describe the measures taken to verify the reproducibility of experimental findings. For each experiment, note whether any attempts to repeat the experiment failed OR state that all attempts to repeat the experiment were successful.* |
| Randomization | *Describe how samples/organisms/participants were allocated into groups. If allocation was not random, describe how covariates were controlled. If this is not relevant to your study, explain why.* |
| Blinding | *Describe the extent of blinding used during data acquisition and analysis. If blinding was not possible, describe why OR explain why blinding was not relevant to your study.* |

Did the study involve field work? ☐ Yes ☐ No

## Field work, collection and transport

| Field conditions | *Describe the study conditions for field work, providing relevant parameters (e.g. temperature, rainfall).* |
| Location | *State the location of the sampling or experiment, providing relevant parameters (e.g. latitude and longitude, elevation, water depth).* |
| Access & import/export | *Describe the efforts you have made to access habitats and to collect and import/export your samples in a responsible manner and in compliance with local, national and international laws, noting any permits that were obtained (give the name of the issuing authority, the date of issue, and any identifying information).* |
| Disturbance | *Describe any disturbance caused by the study and how it was minimized.* |

# Reporting for specific materials, systems and methods

We require information from authors about some types of materials, experimental systems and methods used in many studies. Here, indicate whether each material, system or method listed is relevant to your study. If you are not sure if a list item applies to your research, read the appropriate section before selecting a response.

## Materials & experimental systems

| n/a | Involved in the study |
| --- | --- |
| ☐ | ☐ Antibodies |
| ☐ | ☐ Eukaryotic cell lines |
| ☐ | ☐ Palaeontology and archaeology |
| ☐ | ☐ Animals and other organisms |
| ☐ | ☐ Human research participants |
| ☐ | ☐ Clinical data |
| ☐ | ☐ Dual use research of concern |

## Methods

| n/a | Involved in the study |
| --- | --- |
| ☐ | ☐ ChIP-seq |
| ☐ | ☐ Flow cytometry |
| ☐ | ☐ MRI-based neuroimaging |

## Antibodies

| Antibodies used | *Describe all antibodies used in the study; as applicable, provide supplier name, catalog number, clone name, and lot number.* |
| Validation | *Describe the validation of each primary antibody for the species and application, noting any validation statements on the manufacturer's website, relevant citations, antibody profiles in online databases, or data provided in the manuscript.* |

## Eukaryotic cell lines

Policy information about cell lines

| Cell line source(s) | *State the source of each cell line used.* |
| Authentication | *Describe the authentication procedures for each cell line used OR declare that none of the cell lines used were authenticated.* |
| Mycoplasma contamination | *Confirm that all cell lines tested negative for mycoplasma contamination OR describe the results of the testing for mycoplasma contamination OR declare that the cell lines were not tested for mycoplasma contamination.* |
| Commonly misidentified lines (See ICLAC register) | *Name any commonly misidentified cell lines used in the study and provide a rationale for their use.* |

# Palaeontology and Archaeology

Specimen provenance

*Provide provenance information for specimens and describe permits that were obtained for the work (including the name of the issuing authority, the date of issue, and any identifying information). Permits should encompass collection and, where applicable, export.*

Specimen deposition

*Indicate where the specimens have been deposited to permit free access by other researchers.*

Dating methods

*If new dates are provided, describe how they were obtained (e.g. collection, storage, sample pretreatment and measurement), where they were obtained (i.e. lab name), the calibration program and the protocol for quality assurance OR state that no new dates are provided.*

☐ Tick this box to confirm that the raw and calibrated dates are available in the paper or in Supplementary Information.

Ethics oversight

*Identify the organization(s) that approved or provided guidance on the study protocol, OR state that no ethical approval or guidance was required and explain why not.*

Note that full information on the approval of the study protocol must also be provided in the manuscript.

# Animals and other organisms

Policy information about studies involving animals; ARRIVE guidelines recommended for reporting animal research

Laboratory animals

*For laboratory animals, report species, strain, sex and age OR state that the study did not involve laboratory animals.*

Wild animals

*Provide details on animals observed in or captured in the field; report species, sex and age where possible. Describe how animals were caught and transported and what happened to captive animals after the study (if killed, explain why and describe method; if released, say where and when) OR state that the study did not involve wild animals.*

Field-collected samples

*For laboratory work with field-collected samples, describe all relevant parameters such as housing, maintenance, temperature, photoperiod and end-of-experiment protocol OR state that the study did not involve samples collected from the field.*

Ethics oversight

*Identify the organization(s) that approved or provided guidance on the study protocol, OR state that no ethical approval or guidance was required and explain why not.*

Note that full information on the approval of the study protocol must also be provided in the manuscript.

# Human research participants

Policy information about studies involving human research participants

Population characteristics

*Describe the covariate-relevant population characteristics of the human research participants (e.g. age, gender, genotypic information, past and current diagnosis and treatment categories). If you filled out the behavioural & social sciences study design questions and have nothing to add here, write "See above."*

Recruitment

*Describe how participants were recruited. Outline any potential self-selection bias or other biases that may be present and how these are likely to impact results.*

Ethics oversight

*Identify the organization(s) that approved the study protocol.*

Note that full information on the approval of the study protocol must also be provided in the manuscript.

# Clinical data

Policy information about clinical studies
All manuscripts should comply with the ICMJE guidelines for publication of clinical research and a completed CONSORT checklist must be included with all submissions.

Clinical trial registration

*Provide the trial registration number from ClinicalTrials.gov or an equivalent agency.*

Study protocol

*Note where the full trial protocol can be accessed OR if not available, explain why.*

Data collection

*Describe the settings and locales of data collection, noting the time periods of recruitment and data collection.*

Outcomes

*Describe how you pre-defined primary and secondary outcome measures and how you assessed these measures.*

# Dual use research of concern

Policy information about dual use research of concern

## Hazards

Could the accidental, deliberate or reckless misuse of agents or technologies generated in the work, or the application of information presented in the manuscript, pose a threat to:

No | Yes
☒ | ☐ Public health
☒ | ☐ National security
☒ | ☐ Crops and/or livestock
☒ | ☐ Ecosystems
☒ | ☐ Any other significant area

## Experiments of concern

Does the work involve any of these experiments of concern:

No | Yes
☒ | ☐ Demonstrate how to render a vaccine ineffective
☒ | ☐ Confer resistance to therapeutically useful antibiotics or antiviral agents
☒ | ☐ Enhance the virulence of a pathogen or render a nonpathogen virulent
☒ | ☐ Increase transmissibility of a pathogen
☒ | ☐ Alter the host range of a pathogen
☒ | ☐ Enable evasion of diagnostic/detection modalities
☒ | ☐ Enable the weaponization of a biological agent or toxin
☒ | ☐ Any other potentially harmful combination of experiments and agents

# ChIP-seq

## Data deposition

☐ Confirm that both raw and final processed data have been deposited in a public database such as GEO.

☐ Confirm that you have deposited or provided access to graph files (e.g. BED files) for the called peaks.

Data access links
*May remain private before publication.*

*For "Initial submission" or "Revised version" documents, provide reviewer access links. For your "Final submission" document, provide a link to the deposited data.*

Files in database submission

*Provide a list of all files available in the database submission.*

Genome browser session
(e.g. UCSC)

*Provide a link to an anonymized genome browser session for "Initial submission" and "Revised version" documents only, to enable peer review. Write "no longer applicable" for "Final submission" documents.*

## Methodology

Replicates

*Describe the experimental replicates, specifying number, type and replicate agreement.*

Sequencing depth

*Describe the sequencing depth for each experiment, providing the total number of reads, uniquely mapped reads, length of reads and whether they were paired- or single-end.*

Antibodies

*Describe the antibodies used for the ChIP-seq experiments; as applicable, provide supplier name, catalog number, clone name, and lot number.*

Peak calling parameters

*Specify the command line program and parameters used for read mapping and peak calling, including the ChIP, control and index files used.*

Data quality

*Describe the methods used to ensure data quality in full detail, including how many peaks are at FDR 5% and above 5-fold enrichment.*

Software

*Describe the software used to collect and analyze the ChIP-seq data. For custom code that has been deposited into a community repository, provide accession details.*

# Flow Cytometry

## Plots

Confirm that:

☐ The axis labels state the marker and fluorochrome used (e.g. CD4-FITC).

☐ The axis scales are clearly visible. Include numbers along axes only for bottom left plot of group (a 'group' is an analysis of identical markers).

☐ All plots are contour plots with outliers or pseudocolor plots.

☐ A numerical value for number of cells or percentage (with statistics) is provided.

## Methodology

| | |
|---|---|
| Sample preparation | *Describe the sample preparation, detailing the biological source of the cells and any tissue processing steps used.* |
| Instrument | *Identify the instrument used for data collection, specifying make and model number.* |
| Software | *Describe the software used to collect and analyze the flow cytometry data. For custom code that has been deposited into a community repository, provide accession details.* |
| Cell population abundance | *Describe the abundance of the relevant cell populations within post-sort fractions, providing details on the purity of the samples and how it was determined.* |
| Gating strategy | *Describe the gating strategy used for all relevant experiments, specifying the preliminary FSC/SSC gates of the starting cell population, indicating where boundaries between "positive" and "negative" staining cell populations are defined.* |

☐ Tick this box to confirm that a figure exemplifying the gating strategy is provided in the Supplementary Information.

# Magnetic resonance imaging

## Experimental design

| | |
|---|---|
| Design type | *Indicate task or resting state; event-related or block design.* |
| Design specifications | *Specify the number of blocks, trials or experimental units per session and/or subject, and specify the length of each trial or block (if trials are blocked) and interval between trials.* |
| Behavioral performance measures | *State number and/or type of variables recorded (e.g. correct button press, response time) and what statistics were used to establish that the subjects were performing the task as expected (e.g. mean, range, and/or standard deviation across subjects).* |

## Acquisition

| | |
|---|---|
| Imaging type(s) | *Specify: functional, structural, diffusion, perfusion.* |
| Field strength | *Specify in Tesla* |
| Sequence & imaging parameters | *Specify the pulse sequence type (gradient echo, spin echo, etc.), imaging type (EPI, spiral, etc.), field of view, matrix size, slice thickness, orientation and TE/TR/flip angle.* |
| Area of acquisition | *State whether a whole brain scan was used OR define the area of acquisition, describing how the region was determined.* |

Diffusion MRI     ☐ Used     ☐ Not used

## Preprocessing

| | |
|---|---|
| Preprocessing software | *Provide detail on software version and revision number and on specific parameters (model/functions, brain extraction, segmentation, smoothing kernel size, etc.).* |
| Normalization | *If data were normalized/standardized, describe the approach(es): specify linear or non-linear and define image types used for transformation OR indicate that data were not normalized and explain rationale for lack of normalization.* |
| Normalization template | *Describe the template used for normalization/transformation, specifying subject space or group standardized space (e.g. original Talairach, MNI305, ICBM152) OR indicate that the data were not normalized.* |
| Noise and artifact removal | *Describe your procedure(s) for artifact and structured noise removal, specifying motion parameters, tissue signals and physiological signals (heart rate, respiration).* |

| Volume censoring | *Define your software and/or method and criteria for volume censoring, and state the extent of such censoring.* |

## Statistical modeling & inference

| Model type and settings | *Specify type (mass univariate, multivariate, RSA, predictive, etc.) and describe essential details of the model at the first and second levels (e.g. fixed, random or mixed effects; drift or auto-correlation).* |

| Effect(s) tested | *Define precise effect in terms of the task or stimulus conditions instead of psychological concepts and indicate whether ANOVA or factorial designs were used.* |

Specify type of analysis: ☐ Whole brain ☐ ROI-based ☐ Both

| Statistic type for inference<br>(See Eklund et al. 2016) | *Specify voxel-wise or cluster-wise and report all relevant parameters for cluster-wise methods.* |

| Correction | *Describe the type of correction and how it is obtained for multiple comparisons (e.g. FWE, FDR, permutation or Monte Carlo).* |

## Models & analysis

| n/a | Involved in the study |
|-----|----------------------|
| ☐ | ☐ Functional and/or effective connectivity |
| ☐ | ☐ Graph analysis |
| ☐ | ☐ Multivariate modeling or predictive analysis |

| Functional and/or effective connectivity | *Report the measures of dependence used and the model details (e.g. Pearson correlation, partial correlation, mutual information).* |

| Graph analysis | *Report the dependent variable and connectivity measure, specifying weighted graph or binarized graph, subject- or group-level, and the global and/or node summaries used (e.g. clustering coefficient, efficiency, etc.).* |

| Multivariate modeling and predictive analysis | *Specify independent variables, features extraction and dimension reduction, model, training and evaluation metrics.* |

