## [Peer Review File. · Nature Human Behaviour]

Peer Review Information

Journal: Nature Human Behaviour

Manuscript Title: Life expectancy changes since COVID-19

Corresponding author name(s): Jonas Schöley, José Manuel Aburto and Ridhi Kashyap

Reviewer Comments & Decisions:

Decision Letter, initial version:

31st March 2022

Dear Dr Schöley,

Thank you once again for your manuscript, entitled "Bounce backs amid continued losses: Life expectancy changes since COVID-19," and for your patience during the peer review process.

Your manuscript has now been evaluated by 2 reviewers, whose comments are included at the end of this letter. Although the reviewers find your work to be of interest, they also raise some important concerns. We are interested in the possibility of publishing your study in Nature Human Behaviour, but would like to consider your response to these concerns in the form of a revised manuscript before we make a decision on publication.

To guide the scope of the revisions, the editors discuss the referee reports in detail within the team, including with the chief editor, with a view to (1) identifying key priorities that should be addressed in revision and (2) overruling referee requests that are deemed beyond the scope of the current study.

In the case of your manuscript, we agree with Reviewer 2's request that you update all of your data on estimated mortality in revision, given the time lags that affect official estimates. We would also ask that you provide more detailed analyses of the age-specific mortality rates and their contribution to life expectancy loss.

Both reviewers raise concerns about the strength of your analyses on the association between life expectancy and vaccine uptake. Unless these concerns can be addressed with further analyses, we would ask that you move them to the Supplementary Information, with detailed caveats about the impacts of seasonality on these results, as highlighted by both of our reviewers.

Please do make sure that you respond to and address to all other reviewer comments in full. Do not hesitate to get in touch if you would like to discuss these issues further.

Finally, your revised manuscript must comply fully with our editorial policies and formatting requirements. Failure to do so will result in your manuscript being returned to you, which will delay

its consideration. To assist you in this process, I have attached a checklist that lists all of our requirements. If you have any questions about any of our policies or formatting, please don't hesitate to contact me.

In sum, we invite you to revise your manuscript taking into account all reviewer and editor comments. We are committed to providing a fair and constructive peer-review process. Do not hesitate to contact us if there are specific requests from the reviewers that you believe are technically impossible or unlikely to yield a meaningful outcome.

We hope to receive your revised manuscript within 4-6 weeks. I would be grateful if you could contact us as soon as possible if you foresee difficulties with meeting this target resubmission date.

- Include a "Response to the editors and reviewers" document detailing, point-by-point, how you addressed each editor and referee comment. If no action was taken to address a point, you must provide a compelling argument. When formatting this document, please respond to each reviewer comment individually, including the full text of the reviewer comment verbatim followed by your response to the individual point. This response will be used by the editors to evaluate your revision and sent back to the reviewers along with the revised manuscript.
- Highlight all changes made to your manuscript or provide us with a version that tracks changes.

[REDACTED]

We look forward to seeing the revised manuscript and thank you for the opportunity to review your work. Please do not hesitate to contact me if you have any questions or would like to discuss these revisions further.

Sincerely,

Charlotte Payne

Charlotte Payne, PhD
Senior Editor
Nature Human Behaviour

Reviewer expertise:

Reviewer #1: demography, life expectancy, population projections

Reviewer #2: COVID-19, life expectancy

REVIEWER COMMENTS:

Reviewer #1:

Remarks to the Author:

This paper aims at estimating the loss in life expectancy in 29 countries since 2020. It identifies which countries recovered and investigates the contribution of age-specific mortality rates in the changes in LE. The study found greater loss in Eastern Europe countries and a shift from 80+ in 2020 to younger age group in 2021 in the contribution of this loss. The study also explored the relation between the loss in LE and the vaccination rate as well as the contribution of COVID-19 in the excess of mortality. Although this is not the first paper studying the impact of COVID-19 on life expectancy, it provided updated data with latest mortality estimates. It is also original for the estimation of the contribution of age groups mortality rate in the changes in LE. Most other papers usually focus on very few countries or regions, while this one provided estimates for 29 countries.

For the main objective of the paper (the estimation of the loss in LE and the contribution of age groups), the method used is standard and appropriate. I have however concerns about the method used for some secondary analyses.

The first one is about estimation of the loss that can be attributed to COVID-19 deaths. From my understanding, authors picked gross data provided by national institutes and compiled in the COVerAGE-DB database. Many papers have reported discrepancy and inconsistency in the way countries (or even regions within a country) record causes of death. Therefore, without international standardization, there is probably a big bias in the count of deaths from COVID-19. This comparability issue is barely mentioned in the discussion, and we don't really know at what point it explains the big difference observed among countries. I'm afraid some of the main findings on this question might not be completely valid without further investigation. And this leads to some questionable analysis such as this one on p.15/33:

"While COVID-19 was the largest contributor to 2021 life expectancy deficit in nearly all countries, causes other than COVID-19 explained the majority of the 2021 LE deficit in the Netherlands and Greece. Again, this is in line with evidence from Karlinsky and Kobak who found indirect evidence of substantial under-reporting of deaths due to COVID-19 in the Netherlands during the fall/winter wave in 2021 when comparing excess deaths with COVID-19 deaths."

If there is evidence of a substantial under-reporting of deaths due to COVID-19, then how could the authors conclude that causes other than COVID-19 still explain the majority of the 2021 LE deficit? From my point of view, we don't have enough reliable data to conclude anything.

My second concern is about the analysis of the association of the loss in LE with vaccination uptake. As the authors note, the timing of vaccination vs schedule of excess of death is not taken into

consideration in the correlation. In many places, a large part of the loss in the LE probably happened in Winter/early Spring, while the massive vaccination started massively in late Spring/Summer. Therefore, the correlation with the vaccination rate at the 10th month of the year is probably biased for many countries. I believe analyzing this question properly would require a more sophisticated and relevant method than a simple correlation on raw data (which should probably be a complete independent paper by itself).

Those two issues are probably hard to fix, but at the same time, they are not related to the main objectives of the paper. Therefore, I would suggest to simply remove them from the paper, which would also let more space for analyzing a bit more deeply outcomes related to the contribution of age-specific mortality rates. I indeed believe this is a very interesting topic that could be extended. So far, in the text, results are only contrasted between the 60+ and the less than 60. Those very broad age groups could probably be broken down for a more detailed analysis of the contribution of age groups-specific mortality rate to the loss in LE.

Minor issues:

p.2(/33). The objective of the paper is to assess life expectancy changes since 2020, but already the first sentence of the abstract says there are losses around the world with only a few exceptions. This is not clear whether this sentence is the main finding of the paper or some background information. If this is background information, then the paper looks like repeating an exercise that was already done. I would suggest to emphasize on the added value in the objective or to revise the background information sentence by stating explicitly what was missing in the past research. From my understanding, the main contribution of the paper is to link the loss in life expectancy with changes in mortality by ages group.

p.2. In the abstract, results are reported for the age group 80+ vs -80. In the main text, the contrast is between 60+ vs -60. Be consistent.

p3. "The pandemic increased life expectancy inequalities between countries, as life expectancy losses were higher among countries with lower pre-pandemic life expectancy."

This statement is a bit misleading. First, no low level life-expectancy countries are included in the study. Someone reading the paper quickly might think the pandemic amplified the gap between Africa and Europe, since the list of countries included in the study is not mentioned in the title or in the research context. And it is not so obvious in the abstract neither. Second, the reason of this bigger impact for lower pre-pandemic LE is probably because those countries were hit stronger and had maybe weaker capacity to provide care rather than to their initial level of LE. Indeed, all things being equal (same prevalence and fatality rates), the impact of covid-19 on LE should mathematically be bigger in countries with higher life expectancy. I would suggest to reformulate to avoid misinterpretation.

p.3. "Only a few countries including Denmark, Norway, Finland, Australia, South Korea, Iceland and New Zealand did not experience life expectancy losses"

This is probably only among countries which provide data. Otherwise China, Viet Nam and some other should also probably be included.

p.4. "We examine life expectancy in 2021 in 29 countries, including most of Europe, the USA and Chile". Please explain why you selected those 29 countries (I guess because of data availability, but this is not mentioned).

p.6 When loss in LE are reported, specify (at least at the first mention) what numbers in parentheses stand for. First I thought it was for males and females, then I understood it was CI.

p8. Table 1 is a bit overloaded. It can be simplified, or maybe removed since its relevant information looks already provided by Figures 1 and 2.

p.8/9. If the gender gap in LE is wider, that means the impact of COVID-19 (or at least the excess of deaths) is bigger for males than for females, right? Please be more explicit.

p.10. "In most countries the age group 60–79 contributed the most to the LE deficit in 2021. Exceptions were Scotland and Germany, the former having the largest contributions among ages 40–59, the latter among ages 80+." I think those sentences are in the wrong section, which is about cause of death.

p.14 "Because non-COVID mortality also increased in these ages this may be interpreted as the continuation and worsening of a pre-existing mortality crisis among working age adults"
Or maybe this is because of the under-reporting of death attributed to COVID-19.

p.15 Typo: when vaccines were readily available

Guillaume Marois

Reviewer #2:

Remarks to the Author:

This paper examines life expectancy losses in 29 countries during 2020 and 2021 to understand the heterogeneity of the mortality impact of the COVID-19 pandemic. The main finding is that 2021 saw much more variation across countries than 2020, with some countries experiencing even larger life expectancy losses and others returning to pre-pandemic life expectancy levels. This is an important study that is one of the first to examine the impact of the pandemic in 2021.

The introduction states that “fluctuations in life expectancy are not uncommon.” I think this statement requires more elaboration. Reductions in life expectancy that are even close in magnitude to those during COVID are very uncommon and are typically the result of epidemics or violent conflicts. I would recommend providing context for readers by providing examples of the fluctuations in life expectancy that had rapid bounce backs and how these situations are different from the COVID pandemic.

Data: I see that all data were downloaded on February 18, 2022. Can you provide more information about the level of completeness or comparability of completeness across countries as of this download date? I am only familiar with the US data, which have a very long lag time for all-cause mortality and so any counts obtained are likely an underestimate as of this date. Even if they technically cover deaths through the end of 2021, many deaths get reported and processed at later dates. I would also recommend updating these estimates with the most recently available data when doing the next revision.

I think the associations between vaccination coverage and LE declines is an interesting contribution of this paper. Why was October 1, 2021 chosen as the date of vaccination coverage? Is this due to data limitations? A limitation of this is that this captures coverage relatively late in the year, but some of the deadliest months of 2021 were in the early winter months before many age groups were eligible.

Can the authors provide a brief description of why these countries are included in the short-term mortality fluctuation database? Were any other countries included in the STMF database but excluded from the present analysis and, if so why?

The graphs are amazing: information-rich and easy to understand. I just have a few minor comments to help them better stand on their own:

Figure 1: The “19/21”, “19/20” etc. abbreviations for change in life expectancy imply a division or relative change to me. “19-20” might be more straightforward. There needs to be a definition of “as of week” in the figure caption, but hopefully with updated data all countries can have complete-year data.

Figure 2: Needs more description about how the arrows should be interpreted. Should the total change be the sum of the individual arrows?

Figure 3: Even though it’s obvious, the description should state that blue is increase and red is decrease.

Table 1 is too dense and I found it difficult to extract important findings. It is hard to remember what the solid and hollow triangles represent and which direction means 60+ when looking at the table. If you keep the color scheme, the description needs to note that black/gray means increase and red means decrease and that the shade indicates significance. I think the LE deficit columns could be their own table or figure or be moved to the supplementary information and this would free up space in the table.

Additional limitations:

There is no discussion of differences in the health profiles of these countries. The countries with lower pre-pandemic LE had greater losses, which would support the notion that countries with less healthy populations had worse mortality from COVID.

Low- and middle-income countries are largely excluded. Several "peer" countries of these wealthy countries that presumably have high-quality data are not included, including many that had strict COVID protocols like Japan and Australia. What are the implications of these exclusions for the conclusions?

Author Rebuttal to Initial comments

Revision 9, May 2022

Author responses in blue.

Reviewer 1

This paper aims at estimating the loss in life expectancy in 29 countries since 2020. It identifies which countries recovered and investigates the contribution of age-specific mortality rates in the changes in LE. The study found greater loss in Eastern Europe countries and a shift from 80+ in 2020 to younger age group in 2021 in the contribution of this loss. The study also explored the relation between the loss in LE and the vaccination rate as well as the contribution of COVID-19 in the excess of mortality. Although this is not the first paper studying the impact of COVID-19 on life expectancy, it provided updated data with latest mortality estimates. It is also original for the estimation of the contribution of age groups mortality rate in the changes in LE. Most other papers usually focus on very few countries or regions, while this one provided estimates for 29 countries.

For the main objective of the paper (the estimation of the loss in LE and the contribution of age groups), the method used is standard and appropriate. I have however concerns about the method used for some secondary analyses.

1.1 Under Registration of COVID-19 deaths. *From my understanding, authors picked gross data provided by national institutes and compiled in the COVERAGE-DB database. Many papers have reported discrepancy and inconsistency in the way countries (or even regions within a country) record causes of death. Therefore, without international standardization, there is probably a big bias in the count of deaths from COVID-19. This comparability issue is barely mentioned in the discussion, and we don't really know at what point it explains the big difference observed among countries. I'm afraid some of the main findings on this question might not be completely valid without further investigation. And this leads to some questionable analysis such as this one on p.15/33:*

“While COVID-19 was the largest contributor to 2021 life expectancy deficit in nearly all countries, causes other than COVID-19 explained the majority of the 2021 LE deficit in the Netherlands and Greece. Again, this is in line with evidence from Karlinsky and Kobak who found indirect evidence of substantial under-reporting of deaths due to COVID-19 in the Netherlands during the fall/winter wave in 2021 when comparing excess deaths with COVID-19 deaths.” *If there is evidence of a substantial under-reporting of deaths due to COVID-19, then how could the authors conclude that causes other than COVID-19 still explain the majority of the 2021 LE deficit? From my point of view, we don’t have enough reliable data to conclude anything.*

Thank you for raising this concern and we are in agreement regarding the comparability issue. To facilitate the correct interpretation of our results we added a disclaimer to the main text and directly to the decomposition plots reading:

“Note that COVID-19 related deaths are counted differently across countries and that some cross-country differences will be explained by different reporting conventions as outlined in the discussion.”

We admit where results are inconclusive and discuss the comparability issue with reference to the best available evidence thus far:

“We found lower than expected mortality due to non-COVID-19 causes in 2021 in Belgium, England & Wales, France, and Slovenia (Figure 4). If these results are an artefact of an overly broad definition of COVID-19 related deaths or point towards a genuine decline in non-COVID mortality due to e.g. mortality displacement or the lack of flu-deaths is unclear at this point. For France, over-counting COVID-19 deaths seems unlikely as the underlying data on COVID-19 death counts used here originates from Santé Publique France, which uses a very strict definition of “death due to COVID-19” (Garcia 2021).”

We clarify that the result for the Netherlands, where nominally non-COVID deaths contributed most to the life expectancy losses may be spurious:

“COVID-19 was the largest contributor to the 2021 life expectancy deficit in all analyzed countries, but the Netherlands. This is despite different reporting standards and thus provides additional strong evidence for the direct effect of COVID-19 on increases mortality. The result for the Netherlands may be spurious as Karlinski (Karlinsky 2021) found indirect evidence of substantial under-reporting of deaths due to COVID-19 in the Netherlands during the fall/winter wave in 2021 when comparing excess deaths with COVID-19 deaths. However, because factors other than undercounting can influence the relationship between COVID-19 deaths and excess deaths, more detailed cause of death information is needed to assess the impact of non COVID-19 related mortality on 2021 LE deficits.”

Please note that in the original manuscript we already mention the registration problems in the discussion:

“The inconsistent registration of deaths due to COVID-19 across countries (Garcia 2021) complicates any cause-of-death attribution analysis, including ours.”

The cause of death decomposition analysis has a single purpose: to show evidence that COVID-19 is a direct driver of the life expectancy losses instead of, say, indirect effects of lockdown policies as sometimes claimed. To that end we can show that on a country by country basis, despite the registration differences between countries, COVID-19 deaths almost always contribute the largest share to the LE losses. We explicitly mention the registration definition problem and refer to the relevant literature when explaining our results.

1.2 Vaccination analysis timing dimension. *My second concern is about the analysis of the association of the loss in LE with vaccination uptake. As the authors note, the timing of vaccination vs schedule of excess of death is not taken into consideration in the correlation. In many places, a large part of the loss in the LE probably happened in Winter/early Spring, while the massive vaccination started massively in late Spring/Summer. Therefore, the correlation with the vaccination rate at the 10 th month of the year is probably biased for many countries. I believe analyzing this question properly would require a more sophisticated and relevant method than a simple correlation on raw data (which should probably be a complete independent paper by itself).*

Thank you and we share your concern. In order to eliminate the timing bias we restricted the analysis to the 4th quarter of 2021. Based on weekly death counts, we calculated observed and expected life expectancy over the period October through December 2021. We then calculated the age specific contributions to the life expectancy deficit over this part of the year and plotted them against the share of the population twice vaccinated by October 1st.

Eliminating the vaccination timing bias leads to a strengthening of the inverse relationship between vaccination uptake and life expectancy deficit. Below you see the relationship under the old analysis contrasted with the relationship as we report it now.

Old results for whole year 2021:

New results for Q4 2021:

We agree that the relationship is still confounded by other factors and state so in our discussion, however, we strongly believe that reporting the crude correlation is still a valuable and timely contribution which fits the scope of our paper.

1.3 Discussion of detailed age decomposition. Those two issues are probably hard to fix, but

at the same time, they are not related to the main objectives of the paper. Therefore, I would suggest to simply remove them from the paper, which would also let more space for analyzing a bit more deeply outcomes related to the contribution of age-specific mortality rates. I indeed believe this is a very interesting topic that could be extended. So far, in the text, results are only contrasted between the 60+ and the less than 60. Those very broad age groups could probably be broken down for a more detailed analysis of the contribution of age groups-specific mortality rate to the loss in LE.

Thank you for this suggestion and we understand where you are coming from - this could very well be a paper focussed solely on all cause mortality during the pandemic. But we made a different choice - we want this paper to be broader and we want to contextualise the classical life table estimates with data specific to the COVID-19 pandemic, namely registered COVID-19 deaths and vaccinations. The unique contribution that our group can make is to bring together age-specific data on all-cause deaths, vaccinations, and registered COVID-19 deaths. While the COVID related data is more uncertain than the all cause mortality data, it is the best data we have and it still carries information. The cause-of-death decomposition, despite the bias due to registration differences, is a strong piece of evidence for COVID-19 as the main driver behind the life expectancy losses. As such, it is a vital part of this paper and a safeguard against claims that the life expectancy losses are the indirect result of lockdown measures or similar disinformation. The vaccination analysis is non-causal and likely confounded, and we gladly say so in the discussion, but it is also the first estimate of an expected and vital effect, namely that vaccinations helped to limit the life expectancy deficit experienced during the approaching Delta wave in fall/winter 2021.

We are not pushing surprising results based on uncertain data - we are reporting completely expected results based on the best available data. Our results based on the registered COVID-19 death data and on the vaccination data are completely in line with previous expectations: COVID-19 deaths are a direct driver of the life expectancy losses and vaccinations work to limit the excess mortality and therefore life expectancy deficits. Thus, we see no problem in publishing these results. Of course we notice data limitations in our discussion.

1.4 Abstract. *p.2(/33). The objective of the paper is to assess life expectancy changes since 2020, but already the first sentence of the abstract says there are losses around the world with only a few exceptions. This is not clear whether this sentence is the main finding of the paper or some background information. If this is background information, then the paper looks like repeating an exercise that was already done. I would suggest to emphasize on the added value in the objective or to revise the background information sentence by stating explicitly what was missing in the past research. From my understanding, the main contribution of the paper is to link the loss in life expectancy with changes in mortality by ages group.*

Thank you for this observation. Following your advice, we have rephrased some sentences

in the abstract making sure that the added value is emphasised. You are correct that one of the main contributions of the paper is showing life-expectancy losses and their link with changes in mortality by age groups. Another key contribution of our paper is estimates of life expectancy for 2021 in a comparable cross-country perspective juxtaposed with past trends, including the first year of the pandemic 2020. The abstract now reads:

“The COVID-19 pandemic triggered an unprecedented rise in mortality that translated into life expectancy losses around the world, with only a few exceptions. We estimate life expectancy changes in 29 countries since 2020, including most of Europe, the US and Chile, attribute them to mortality changes by age group, and compare them to historic life expectancy shocks. Our results show divergence in mortality impacts of the pandemic in 2021. While countries in Western Europe experienced bounce-backs from life expectancy losses of 2020, Eastern Europe and the US witnessed sustained and substantial life expectancy deficits. Life expectancy deficits among ages 60+ were strongly correlated with measures of vaccination uptake. In contrast to 2020, the age profile of excess mortality in 2021 was younger with those in under-80 age groups contributing more to life expectancy losses. However, even in 2021, registered COVID-19 deaths continued to account for most life expectancy losses.”

1.5 Consistency of age grouping. p.2. *In the abstract, results are reported for the age group 80+ vs -80. In the main text, the contrast is between 60+ vs -60. Be consistent.*

We have revised the sentence in the abstract, it now reads:

“In contrast to 2020, the age profile of excess mortality in 2021 was younger with those in under-60 age groups contributing substantially to life expectancy losses. However, even in 2021, registered COVID-19 deaths continued to account for most life expectancy losses.”

1.6 The pandemics effect on life expectancy convergence. p3. “The pandemic increased life expectancy inequalities between countries, as life expectancy losses were higher among countries with lower pre-pandemic life expectancy.” *This statement is a bit misleading. First, no low level life-expectancy countries are included in the study. Someone reading the paper quickly might think the pandemic amplified the gap between Africa and Europe, since the list of countries included in the study is not mentioned in the title or in the research context. And it is not so obvious in the abstract neither. Second, the reason of this bigger impact for lower pre-pandemic LE is probably because those countries were hit stronger and had maybe weaker capacity to provide care rather than to their initial level of LE. Indeed, all things being equal (same prevalence and fatality rates), the impact of covid-19 on LE should mathematically be bigger in countries with higher life expectancy. I would suggest to reformulate to avoid misinterpretation.*

Thank you for this observation. We agree that some statements about our results needed clarification about the countries included in our study, which as you pointed out, are mostly high-income low-mortality countries. This is noted in the abstract as we mention that we cover most Europe, the USA and Chile explicitly. We, however, have clarified these statements throughout the manuscript, for example the sentence now reads:

“The pandemic increased life expectancy inequalities between the 29 low-mortality countries that we analyze, as life expectancy losses were higher among countries with lower pre-pandemic life expectancy.”

Regarding your second point, the manuscript already included a sentence pointing out that cross-country differences in 2021 LE deficits were bigger in countries with pre-pandemic lower LE.

1.7 Countries without life expectancy losses. p.3. “Only a few countries including Denmark, Norway, Finland, Australia, South Korea, Iceland and New Zealand did not experience life expectancy losses” *This is probably only among countries which provide data. Otherwise China, Viet Nam and some other should also probably be included.*

We changed the sentence to:

“Among those countries with publicly available data on deaths, only a few, including Denmark, Norway, Finland, Australia, South Korea, Iceland and New Zealand did not experience life expectancy losses.”

1.8 Selection of countries. p.4. “We examine life expectancy in 2021 in 29 countries, including most of Europe, the USA and Chile”. *Please explain why you selected those 29 countries (I guess because of data availability, but this is not mentioned).*

See also 2.4. The STMF database includes a total of 38 countries but for some of these countries the age grouping of death counts is too coarse to reliably estimate life expectancy. A case in point is New Zealand with only three broad age groups. The 29 countries we selected have small enough age groups for us to reliably estimate life expectancy and have complete data for years 2020 and 2021.

We added a clarifying statement to the methods section:

“The 38 countries represented in the database have been selected based on completeness of their death registration and census data. To allow for a reliable estimation of life expectancy changes and their age contributions, we further selected those 29 countries for which death counts for 2020 and 2021 were reported across at least 10 distinct age groups.”

1.9 Clearly indicate CI's. p.6 *When loss in LE are reported, specify (at least at the first mention) what numbers in parentheses stand for. First I thought it was for males and females, then I understood it was CI.*

Thank you for this observation. We changed the beginning of the paragraph, so the meaning of the used CI gets clearer. The first sentence of the paragraph now reads:

“Bulgaria experienced 17.8 months of LE decline in 2020 with a CI from 16.3 to 19.3 months.”

1.10 Simplify Table 1. p8. *Table 1 is a bit overloaded. It can be simplified, or maybe removed since its relevant information looks already provided by Figures 1 and 2.*

See also 2.9. To reduce visual distractions we limited the color coding to the symbols, dropped the shading for non-significant values, and wrote the uncertainty intervals in a more compact fashion. Based on feedback on our last publication on COVID-19 life expectancy losses, we provide the table for the readers who prefer tables over figures.

- 1.11 Clarify gender gap direction.** p.8/9. *If the gender gap in LE is wider, that means the impact of COVID-19 (or at least the excess of deaths) is bigger for males than for females, right? Please be more explicit.*

We agree that the interpretation of the changes in the gap between female and male LE needed further explanation. We added the following sentence to the paragraph:

“This finding indicates that for most countries males were more affected by excess death.”

- 1.12 Wrong placement of paragraph.** p.10. *“In most countries the age group 60-79 contributed the most to the LE deficit in 2021. Exceptions were Scotland and Germany, the former having the largest contributions among ages 40-59, the latter among ages 80+.” I think those sentences are in the wrong section, which is about cause of death.*

The paragraph is correctly placed, but the section title was misleading. We renamed it to:

“Life expectancy deficit contributions by cause of death and age”

Note that we analysed both the age contributions to the life expectancy deficit, and the age contributions to the life expectancy changes.

- 1.13 p.14 COVID-19 under-registration** “Because non-COVID mortality also increased in these ages this may be interpreted as the continuation and worsening of a pre-existing mortality crisis among working age adults” Or maybe this is because of the under-reporting of death attributed to COVID-19.

Thank you. That may be part of the explanation which we should mention. We added the sentence:

“However, part of the effect may be due to under-registration of COVID-19 deaths among the working age population.”

- 1.14 Typo** p.15 *Typo: when vaccines were readily available*

Thank you for this observation. We corrected the typo in the manuscript.

Reviewer 2

This paper examines life expectancy losses in 29 countries during 2020 and 2021 to understand the heterogeneity of the mortality impact of the COVID-19 pandemic. The main finding is that 2021 saw much more variation across countries than 2020, with some

countries experiencing even larger life expectancy losses and others returning to pre-pandemic life expectancy levels. This is an important study that is one of the first to examine the impact of the pandemic in 2021.

2.1 Fluctuations in life expectancy. *The introduction states that “fluctuations in life expectancy are not uncommon.” I think this statement requires more elaboration. Reductions in life expectancy that are even close in magnitude to those during COVID are very uncommon and are typically the result of epidemics or violent conflicts. I would recommend providing context for readers by providing examples of the fluctuations in life expectancy that had rapid bounce backs and how these situations are different from the COVID pandemic.*

Thank you. We rephrased the sentence so that there can be no doubt about the magnitude of the COVID-19 induced mortality shock if this paragraph is read in isolation. The new sentence states:

“In contrast with these short-term fluctuations, however, the COVID-19 pandemic induced global and severe mortality shocks in 2020 and, as of spring 2022, is still ongoing.”

We are in complete agreement about the need for historical examples of mortality shocks and fluctuations and thus dedicated a whole section to this aspect. In the section labelled “Comparison with past mortality shocks” we look back at more than 100 years of life expectancy changes across a range of countries and historically contextualise the mortality impact of the current COVID-19 pandemic. Figure 6 clearly indicates the unusual severity and scope of the COVID-19 induced mortality crisis.

2.2 Data quality. *Data: I see that all data were downloaded on February 18, 2022. Can you provide more information about the level of completeness or comparability of completeness across countries as of this download date? I am only familiar with the US data, which have a very long lag time for all-cause mortality and so any counts obtained are likely an underestimate as of this date. Even if they technically cover deaths through the end of 2021, many deaths get reported and processed at later dates. I would also recommend updating these estimates with the most recently available data when doing the next revision.*

The concern of under-registration is also very much on our minds. To that end, we check our results weekly with updated data. Around mid February 2022 the annual life expectancy losses and other results in our paper stabilised. The rankings between countries only changed slightly (not among the top) as did the numerical results. This is when we decided that death registration is complete enough to publish the paper on medRxiv. Today, end of April 2022, we’ve updated the data yet again and the results are very close to the results you have reviewed, which gives us great confidence that late registration won’t revert our results or conclusions in the future.

However, more can be done: We calculated completeness of registration ratios for the

countries in our paper based on the World Mortality Dataset (WMD). The WMD aggregates weekly or monthly total death counts for a number of countries from official sources and uploads this data to github. Because the data is updated daily and github provides an archive of the data stretching back to 2021 we were able to calculate the completeness ratio $P = \text{Total deaths in 2020 as reported by April 26, 2021} / \text{Total deaths in 2020 as reported by April 30, 2022}$.

For the U.S., by April 26, 2021, 99.82% of all deaths in 2020 have been registered, if we assume death registration to be 100% by April 30, 2022. Further assuming that death registration for 2021 is at least as fast as the death registration in 2020, this yields less than 0.2% of all 2021 deaths missing from our U.S. data. The impact on the life expectancy loss estimates is minor and likely on the scale of weeks.

Looking beyond the U.S., under the method outlined above, the worst under-registration in our data is with Finland at 1.02% of 2021 deaths still missing. Accounting for these missing deaths will not change the conclusion that Finland is among the best performing countries in terms of life expectancy losses during 2020 and 2021. For most of the countries reported here, missing deaths due to late registration are likely less than 0.1%.

We updated all results with data from April 26, 2022.

2.3 Vaccination analysis timing dimension. *I think the associations between vaccination coverage and LE declines is an interesting contribution of this paper. Why was October 1, 2021 chosen as the date of vaccination coverage? Is this due to data limitations? A limitation of this is that this captures coverage relatively late in the year, but some of the deadliest months of 2021 were in the early winter months before many age groups were eligible.*

Thank you. In response to comment 1.2 we calculated the life expectancy deficit for the vaccination analysis only for October through December 2021. In this way, the share of people vaccinated by October 1st is aligned in time with the life expectancy deficit we calculate. We choose October 1st and the fourth quarter of 2021 because here, the effect of vaccination differences between countries is most clearly visible: During the first quarter of the year vaccination quotas were still too low to have a marked effect on cross-country differences. Over spring and summer 2021 seasonality effects reduced the severity of the COVID-19 pandemic and associated excess deaths in many countries. In the 4th quarter the vaccines were crucial however with large differences in vaccination quotas between countries and the emergence of the Delta variant.

2.4 Selection of countries. *Can the authors provide a brief description of why these countries are included in the short-term mortality fluctuation database? Were any other countries included in the STMF database but excluded from the present analysis and, if so why?*

See also 1.8. The STMF database is co-hosted by the Max Planck Institute for

Demographic Research, Berkeley, and the INED Institute in Paris. The database authors state that the “database is limited by design to populations where death registration and census data are virtually complete”. The data they publish is based on raw data provided by collaborating statistical offices and further harmonised across time and space to facilitate international comparisons. Thus, what we can provide based on the STMF data is an international snapshot of extremely high quality life expectancy estimates. Without doubt, other groups will provide life expectancy estimates for the whole globe in the near future. Such estimates will be based on indirect estimates and form a valuable albeit uncertain contribution. We think it is best to first publish what is known with certainty.

The STMF database includes a total of 38 countries but for some of these countries the age grouping of death counts is too coarse to reliably estimate life expectancy. A case in point is New Zealand with only three broad age groups. The 29 countries we selected have small enough age groups for us to reliably estimate life expectancy and have complete data for years 2020 and 2021.

We added a clarifying statement to the methods section:

“The 38 countries represented in the database have been selected based on completeness of their death registration and census data. To allow for a reliable estimation of life expectancy changes and their age contributions, we further selected those 28 countries for which death counts for 2020 and 2021 were reported across at least 10 distinct age groups.”

2.5 *Graphs.* *The graphs are amazing: information-rich and easy to understand. I just have a few minor comments to help them better stand on their own:*

Thank you very much.

2.6 *Period labelling.* *Figure 1: The “19/21”, “19/20” etc. abbreviations for change in life expectancy imply a division or relative change to me. “19-20” might be more straightforward. There needs to be a definition of “as of week” in the figure caption, but hopefully with updated data all countries can have complete-year data.*

Agreed. We have made the changes accordingly throughout the manuscript.

2.7 *Add description to Figure 2.* *Figure 2: Needs more description about how the arrows should be interpreted. Should the total change be the sum of the individual arrows?*

Done. The new caption reads:

“Age contributions to life expectancy changes since 2019 separated for 2020 and 2021. The position of the arrowhead indicates the total contribution of mortality changes in a given age group to the change in life expectancy at birth since 2019. The discontinuity in the arrow indicates those contributions separately for the years 2020 and 2021. Annual contributions can

compound or reverse. The total life expectancy change from 2019 to 2021 in a given country is the sum of the arrowhead positions across age.”

2.8 Describe the colors in Figure 3. *Figure 3: Even though it’s obvious, the description should state that blue is increase and red is decrease.*

Done. The new caption reads:

“Change in the female life expectancy advantage from 2019 through 2021. Blue colors indicate an increase and red colors a decrease in the female life expectancy advantage. Muted colors indicate non-significant changes.”

2.9 Simplify Table 1. *Table 1 is too dense and I found it difficult to extract important findings. It is hard to remember what the solid and hollow triangles represent and which direction means 60+ when looking at the table. If you keep the color scheme, the description needs to note that black/gray means increase and red means decrease and that the shade indicates significance. I think the LE deficit columns could be their own table or figure or be moved to the supplementary information and this would free up space in the table.*

See also 1.10. Yes, you are right. To reduce visual distractions we limited the color coding to the symbols, dropped the shading for non-significant values, and wrote the uncertainty intervals in a more compact fashion. We’ve added a description of the color scheme to the table notes.

2.10 Discuss health profile differences. *There is no discussion of differences in the health profiles of these countries. The countries with lower pre-pandemic LE had greater losses, which would support the notion that countries with less healthy populations had worse mortality from COVID.*

We expanded a paragraph in the discussion which now reads:

Pre-pandemic differences in underlying conditions such as obesity and diabetes also may also have contributed to an increased mortality burden in working age USA adults compared to European counterparts (Pongiglione 2022). Regarding global comparisons, the evidence for co-morbidity prevalence as an important predictor of regional COVID-19 mortality is still weak (Thakur 2021).

We hesitate to speculate much more about this topic because of the weak evidence so far. While there’s strong evidence for the importance of co-morbidities in individual risk assessment, the evidence is weak with respect to health profiles explaining cross-country differences in population level COVID-19 mortality. This effect may be masked by unmeasured treatment differences, but we don’t know yet.

2.11 Scope of our conclusions. *Low- and middle-income countries are largely excluded. Several “peer” countries of these wealthy countries that presumably have high-quality*

data are not included, including many that had strict COVID protocols like Japan and Australia. What are the implications of these exclusions for the conclusions?

Thank you. Yes, that was clearly missing. We added a paragraph to the Discussion section highlighting the selective nature of our data and putting our result in context with indirect life expectancy estimates based on excess deaths which have been published for a larger set of countries. The new paragraph reads:

“As the calculation of life expectancy losses requires timely data on population and death counts by age, we are limited in our analysis to those countries with a reliable vital statistics registration system. Consequently our international analysis based on 29 high and middle income countries may give a skewed impression of the global impact of COVID-19 on life expectancy. Indirect life expectancy estimates for 2020-21 based upon excess deaths (Heuveline 2022) indicate substantial losses across South America, which match or exceed the losses we estimated for Eastern Europe. India and select countries in the Middle East likely had losses on par with the U.S. whereas Russia and Mexico suffered life expectancy losses in excess of the 42 months we estimated for Bulgaria. Little is known about the African continent, due to a lack of reliable death registration, or China, due to restrictive access to the data. Putting our results in context with these indirect estimates: while the life expectancy losses in Central, Western and Northern Europe over the first two years of the pandemic have been drastic given the long trend of declining mortality, they likely have been low when compared across the globe.”

Decision Letter, first revision:

Our ref: NATHUMBEHAV-22030515A

30th May 2022

Dear Dr. Schöley,

Thank you for submitting your revised manuscript "Life expectancy changes since COVID-19 are marked by bounce backs amid continued losses" (NATHUMBEHAV-22030515A). It has now been seen by the original referees and their comments are below. As you can see, the reviewers find that the paper has improved in revision. We will therefore be happy in principle to publish it in Nature Human Behaviour, pending minor revisions to comply with our editorial and formatting guidelines.

We are now performing detailed checks on your paper and will send you a checklist detailing our editorial and formatting requirements within a week. Please do not upload the final materials and make any revisions until you receive this additional information from us.

Sincerely,

Charlotte Payne

Charlotte Payne, PhD
Senior Editor
Nature Human Behaviour

Reviewer #1 (Remarks to the Author):

I'm satisfied with the revision and explanations made by authors

Reviewer #1 (Remarks to the Author):

[None]

Final Decision Letter:

Dear Dr Schöley,

We are pleased to inform you that your Article "Life expectancy changes since COVID-19", has now been accepted for publication in Nature Human Behaviour.

Please note that *Nature Human Behaviour* is a Transformative Journal (TJ). Authors whose manuscript was submitted on or after January 1st, 2021, may publish their research with us through the traditional subscription access route or make their paper immediately open access through payment of an article-processing charge (APC). Authors will not be required to make a final decision about access to their article until it has been accepted. IMPORTANT NOTE: Articles submitted before January 1st, 2021, are not eligible for Open Access publication. Find out more about Transformative Journals

With best regards,

Charlotte Payne

Charlotte Payne, PhD
Senior Editor
Nature Human Behaviour